# Deoxygenation dynamics on the western Nile deep-sea fan during sapropel S1 from seasonal to millennial time-scales

Cécile L. Blanchet[1], Rik Tjallingii[1], Anja M. Schleicher[2], Stefan Schouten[3,4], Martin Frank[5], Achim Brauer[1]

[1] GFZ German Research Centre for Geosciences Potsdam, Section of Climate and Landscape Dynamics, Telegrafenberg, 14743 Potsdam, Germany

[2] GFZ German Research Centre for Geosciences Potsdam, Section of Inorganic and Isotope Geochemistry, Telegrafenberg, 14743 Potsdam, Germany

[3] Royal NIOZ, Department of Marine Microbiology and Biogeochemistry, Landsdiep 4, 1797 SZ 't Horntje, Texel, The Netherlands

[4] University of Utrecht, Department of Earth Sciences, Utrecht, the Netherlands.

[5] GEOMAR Helmholtz Centre for Ocean Research Kiel, Research Unit Paleoceanography, Wischhofstrasse 1-3, D-24148 Kiel, Germany

*Correspondence to*: Cécile L. Blanchet (blanchet@gfz-potsdam.de)

**Abstract**

Ocean deoxygenation is a rising threat to marine ecosystems and food resources under present climate warming
conditions. Organic-rich sapropel layers deposited in the Mediterranean Sea provide a natural laboratory to study
the processes that have controlled changes in seawater oxygen levels in the recent geological past. Our study is
based on three sediment cores spanning the last 10,000 years and located on a bathymetric transect offshore the
western distributaries of the Nile delta. These cores are partly to continuously laminated in the sections recording
sapropel S1, which is indicative of bottom-water anoxia above the western Nile deep-sea fan. We used a
combination of microfacies analyses and inorganic and organic geochemical measurements to reconstruct changes
in oxygenation conditions at seasonal to millennial time-scales. Millimetre-thick laminations are composed of
detrital, biogenic and chemogenic sublayers reflecting seasonal successions of sedimentation. Dark layers reflect
the deposition of summer floods and two types of light layers correspond to autumn plankton blooms and authigenic
carbonates formed in the water-column during spring-early summer. The isotopic signature of these carbonates
suggests permanent anoxic to euxinic bottom waters resulting in high levels of anaerobic remineralisation of
organic matter and highlights their potential to reconstruct seawater chemistry at times when benthic fauna was
absent. Ratios of major elements combined to biomarkers of terrestrial and marine organic matter and redox-
sensitive compounds allow to track changes in terrigenous input, primary productivity and past deoxygenation
dynamics on millennial time-scales. Rapid fluctuations of oxygenation conditions in the upper 700 m water depth
occurred above the Nile deep-sea fan between 10 and 6.5 ka BP while deeper cores recorded more stable anoxic
conditions. Synchronous changes in terrigenous input, primary productivity and past oxygenation dynamics after
6.5 ka BP show that runoff-driven eutrophication played a central role in rapid oxygenation changes in the south-
eastern Levantine Basin. These findings are further supported by other regional records and reveal time-
transgressive changes in oxygenation state driven by rapid changes in primary productivity during a period of long-
term deep-water stagnation.

**1 Introduction**

The present-day Mediterranean Sea is well-oxygenated due to a vigorous thermohaline circulation of intermediate-
and deep-water initiated in the eastern part of the basin (Pinardi et al., 2015; Roether et al., 1996). However,
recurring episodes of water-column stratification and severe oxygen depletion have been recorded in Mediterranean
sediments (Cramp and O'Sullivan, 1999; Rossignol-Strick, 1985). These episodes marked by sapropel deposits
were caused by drastic modifications of the sea-surface hydrological balance (Rohling, 1994). Orbital-forced
enhancement of monsoonal precipitation over North Africa led to high freshwater discharge into the Eastern
Mediterranean (mostly through the Nile River) and generated large-scale perturbations of the Mediterranean
thermohaline circulation (Rohling et al., 2015). Seawater freshening is also a well-known perturbator in the North
Atlantic that has led to slowing down or even complete shutting down of Atlantic Meridional Overturning
Circulation during the last glacial period (Ganopolski and Rahmstorf, 2001). In the Mediterranean Sea, the input
of relatively low-saline Atlantic seawater also exerted a large control on deep-water ventilation (Rogerson et al.,
2012). The inflow of Atlantic-derived seawater at the Gibraltar Strait as early as 16-17 ka BP during deglacial sea-
level rise increased the buoyancy of surface waters and led to long-lasting deep-water stagnation in the eastern
Mediterranean (Rogerson et al., 2010), which is seen as a pre-condition for the development of basin-scale anoxia
during sapropel S1 (11-6 ka BP) (Cornuault et al., 2018; Grimm et al., 2015).
A specificity of the eastern Mediterranean lies in its current nutrient-limitation and the consequently low primary
productivity (Pujo-Pay et al., 2011). Nutrient-poor surface waters are advected from the western to the eastern
Mediterranean and riverine nutrients (e.g. discharged by the Nile River) are quickly utilized or extracted by
intermediate water circulation flowing westward (Fig. 1a). Prior to the construction of the Aswan Dam in 1965, the
surface waters in the Levantine Basin were seasonally fertilised by riverine nutrient input during the summer Nile
floods (Halim et al., 1967). These so-called "Nile Blooms" of phytoplankton sustained traditional fisheries of
sardines and prawns (Nixon, 2003). After 1965, the Nile River discharge was reduced by 90% and thereby largely
annihilated the main nutrient source to the eastern Mediterranean, initiating the present-day ultra-oligotrophic
conditions. Since the 1980s however, the transition from a flood-sustained to a fertiliser-sustained agriculture in
the Nile valley has led to a larger release of nutrients by the Nile River, which boosted primary productivity and
renewed the fish stocks (Nixon, 2003). Anthropogenic fertilisation processes in coastal waters might have potential
adverse effects as well, such as water deoxygenation or harmful algal blooms (Nixon, 2003). Eutrophication and
severe anoxia due to anthropogenic fertilisation have been indeed observed in semi-enclosed settings such as the
Baltic Sea (Jilbert and Slomp, 2013). In this context of increasing nutrient loading, it is crucial to estimate the
changes in intermediate and deep-water ventilation caused by climate change. Modelling experiments predict a
temperature and salinity rise for the Mediterranean over the next decades, which overall will result in a weakening
or a change in configuration of the thermohaline circulation in the Mediterranean (e.g., Adloff et al., 2015).
Sensitivity simulations showed that reduced deep ventilation leading to basin-wide hypoxia might occur in the
Mediterranean over long time scales (>1000 yr), but is unlikely over shorter time scales (ca. 100 yr) (Powley et al.,
2016). However, the predicted changes in $O_2$ distributions do not account for additional factors that may affect $O_2$
consumption rates, in particular, rising inputs of anthropogenic nutrients or oxygen loss due to rising sea
temperature (i.e., decreased oxygen solubility in warmer waters) (Keeling et al., 2009; Powley et al., 2016).
Sapropels provide a natural laboratory to investigate the complex interplay between oceanographic, hydrological
and biogeochemical processes driving Mediterranean deoxygenation (Rohling et al., 2015; Rossignol-Strick et al.,
1982). Here we present a set of unique sediment cores retrieved along a bathymetric section near the main western
Nile tributary, the Rosetta channel, to investigate the relationships between terrigenous inputs, primary
productivity, and the development of sub- to anoxic conditions in intermediate water masses. The location of our
study site close to the mouth of the Nile River allows to reconstruct changes in fluvial nutrient supply and associated
fertilisation of the marine ecosystem.
**2 Regional context**
Surface waters originating from the North Atlantic flow into the Levantine Basin (Pinardi et al., 2015), which
gradually become more saline due to evaporation as they flow eastwards in the Mediterranean Sea (Fig. 1a).
Elevated salinity of surface waters in the eastern Levantine Basin (ca. 39 PSU) (MEDAR Group, 2002) in
combination with cooler temperatures in winter lead to high surface water density and formation of the Levantine
intermediate water (LIW), which flows westwards at water depths between 200 and 600 m (Cornuault et al., 2018;
Pinardi et al., 2015). Deeper water masses flow below the LIW and form by incorporation of cold surface waters
from the Adriatic Sea and sometimes dense waters from the Aegean Sea to the LIW (Cornuault et al., 2018;
Robinson et al., 1992; Roether et al., 2007). According to recent results, deep convection is also occurring in the
northern Levantine Basin, resulting in the formation of Levantine deep water (LDW) that ventilates the whole
Levantine Basin (Kubin et al., 2019). At present, the LDW bathing the core sites is well oxygenated due to high
rates of LDW formation and low primary productivity (Powley et al., 2016).
The main source of freshwater to the Levantine Basin is the Nile River runoff through the active Rosetta channel
located in the western part of the Nile Delta (Fig. 1a, b). Although the present Nile flow is drastically reduced
compared to historical times (Halim et al., 1967), runoff still leads to marked salinity gradients (halocline) in the
upper 200 m water-depth (w-d), in particular during the summer monsoon (Fig. 1a). The plume of Nile-derived
freshwater is deflected to the East by the long-shore surface current called "Levantine Jet" (Fig. 1b) (Pinardi et al.,

103  2015).


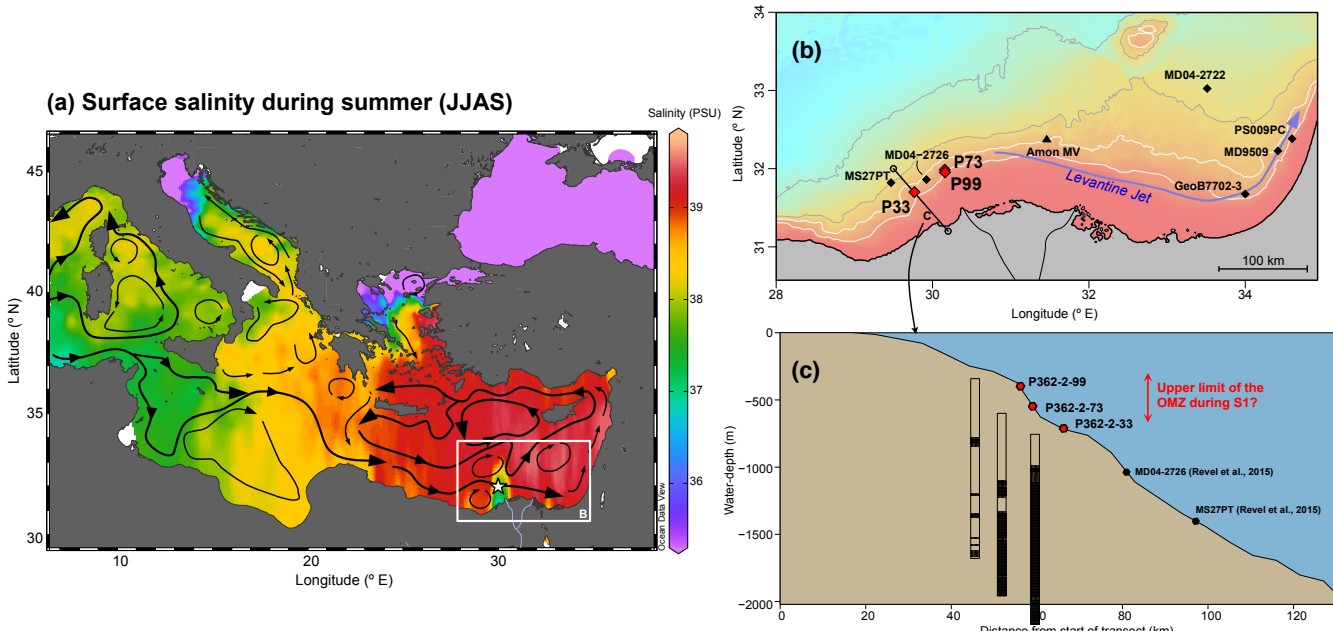

**Figure 1. Regional context and location of the cores. (a) Sea-surface salinity during the summer months (June to September)**
**(MedAtlas, MEDAR Group, 2002) plotted with the Ocean Data View software (Schlitzer, 2018). White star: location of the Nile**
**River freshwater plume. Black arrows: surface currents ((Pinardi et al., 2015; Rohling et al., 2015). (b) Zoom on the western Nile**
**deep-sea fan (DSF), with locations of cores P33, P73 and P99 (this study), as well as neighbouring cores MS27PT and MD04-2726**
**(Revel et al., 2015), MD04-2722 (Tachikawa et al., 2015), GeoB7702-3 (Castañeda et al., 2010), PS009PC (Hennekam et al., 2014),**
**MD9509 (Matthews et al., 2017) (black diamonds) and the Amon mud volcanoe (black triangle). White contours: 500-m and 1000-**
**m isobaths and grey contours: 1500-m and 2000-m isobaths. (c) Bathymetric transect across the western Nile DSF with sediment**
**lithological features (black: laminations and stripes: faint laminations). Map and bathymetric transect in (a) and (b) plotted using**
**the R freeware (R Core Team, 2013), the marmap package (Pante and Simon-Bouhet, 2013) and gridded bathymetric data from**
**Gebco (Gebco team, 2014).**

## 116 3 Material and methods

### 117 3.1 Sediment cores

**Table 1. Sediment cores used for the present study, with names used throughout the paper, water-depths and coordinates.**

| Core Name | Shortened | Water-depth (m) | Latitude (dec. deg.) | Longitude (dec. deg) |
|-----------|-----------|-----------------|----------------------|----------------------|
| P362/2-33 | P33 | 738 | 31.700 | 29.768 |
| P362/2-73 | P73 | 569 | 31.987 | 30.156 |
| P362/2-99 | P99 | 396 | 31.953 | 30.160 |


For our study, we use three gravity cores that were retrieved during the oceanographic campaign P362/2 "West
Nile Delta" onboard the R.V. Poseidon in August 2008 (Feseker et al., 2010) (Table 1). These sediment cores are
located along a bathymetric section on the western Nile deep-sea fan (DSF), in the vicinity of the North Alex and
Giza mud volcanoes (Feseker et al., 2010) (Fig. 1b,c). Cores P362/2-73 (P73) and P362/2-99 (P99) are located

close to each other and ca. 75 km eastward of core P362/2-33 (P33), on the opposite side of the submarine Rosetta canyon (Fig. 1b). Core P33 was recovered at 738 m water-depth, is 5.6 m-long and contains well-preserved mm-scale laminations in the lower 5 m of the core (see Blanchet et al., 2013, 2014, for detailed descriptions) (Fig. 2a). Core P73 was collected at 569 m depth, is 5.4 m-long and contains an alternation of finely laminated (similar to P33) and bioturbated intervals (Fig. 2b). Core P99 is located at 396 m depth, is 5.3 m-long and contains a few laminated intervals within otherwise bioturbated sediments (Fig. 2c). All cores are composed of clay-rich dark-brown hemipelagic mud, which shows significantly lighter colours in the upper 1m of the cores. In cores P33 and P73, several harder carbonate-rich layers were identified (Fig. 2a, b). A quantitative assessment of the number of benthic foraminifera per gram of sediment was realised in the lower part of core P33 (between 75 and 558 cm).

**Table 2: Radiocarbon ages, tie-points and modelled ages for cores P33, P73 and P99. Radiocarbon ages for core P33 from Blanchet et al. (2013), with new modelled ages with uncertainties for the entire core (using Bacon V2.3, Blaauw and Christen, 2011). Six tie points were defined on the Ti/Ca record of P33 (T1 to T6, see sect. 3.2 and Fig. 2). New radiocarbon measurements, tie-points and modelled ages with uncertainties for cores P73 and P99.**

| Sample name | Depth (cm) | material | 14C age ± 1σ | C cont. (mg) | DeltaR ± 1σ | Calib curve | cal. age BP ± 2σ | modelled median age (Bacon) |
|---|---|---|---|---|---|---|---|---|
| **P362/2-33 (P33)** | | | | | | | | |
| | 1.5±1.5 | Steamboat cinders | | | | | 75±75 | 161± 210 |
| KIA 38572 | 30±0.5 | Pk foraminifera | 3825±30 | 1.3 | 0 | Marine13 | 3775±100 | 3840±180 |
| KIA 38573 | 50±0.5 | Pk foraminifera | 5695±30 | 1.3 | 0 | Marine13 | 6110±100 | 6140±150 |
| KIA 38574 | 75±0.5 | Pk foraminifera | 6855±35 | 1.6 | 150±30 | Marine13 | 7235±105 | 7260±180 |
| KIA 37800 | 100±0.5 | Pk foraminifera | 7340±45 | 1.2 | 150±30 | Marine13 | 7655±110 | 7825±150 |
| KIA 38575 | 152±2.5 | Pk foraminifera | 7920±45 | 1.3 | 150±30 | Marine13 | 8240±120 | 8165±130 |
| KIA 37801 | 200±0.5 | Pk foraminifera | 8115±55 | 1.1 | 150±30 | Marine13 | 8430±125 | 8405±120 |
| KIA 38576 | 252±2.5 | Pk foraminifera | 8360±40 | 1.1 | 150±30 | Marine13 | 8725±180 | 8610±115 |
| KIA 37802 | 300±0.5 | Pk foraminifera | 8375±60 | 0.9 | 150±30 | Marine13 | 8750±200 | 8765±110 |
| KIA 38577 | 327±3 | Pk foraminifera | 8495±35 | 1.3 | 150±30 | Marine13 | 8925±160 | 8850±110 |
| KIA 37803 | 399±1 | Pk foraminifera | 8440±45 | 1.4 | 150±30 | Marine13 | 8845±175 | 9050±105 |
| KIA 38578 | 463±2 | Pk foraminifera | 8680±40 | 1 | 150±30 | Marine13 | 9165±150 | 9280±100 |
| KIA 37804 | 499±1 | Pk foraminifera | 8805±45 | 1.5 | 150±30 | Marine13 | 9330±145 | 9410±100 |
| KIA 38579 | 547±3 | Pk foraminifera | 9010±35 | 1.2 | 150±30 | Marine13 | 9515±100 | 9595±125 |
| KIA 37805 | 557±1 | Pk foraminifera | 8780±50 | 0.7 | 150±30 | Marine13 | 9300±160 | 9635±135 |
| T1 | 8 | | | | | | | 1000±440 |

| | | | | | | | | |
|---|---|---|---|---|---|---|---|---|
| T2 | 17.5 | | | | | | | 2250±520 |
| T3 | 31.5 | | | | | | | 4030±150 |
| T4 | 48 | | | | | | | 5965±140 |
| T5 | 80.5 | | | | | | | 7385±110 |
| **P362/2-73 (P73)** | | | | | | | | |
| Poz-113950 | 15 ±1 | Pk foraminifera + pteropod | 505±30 | 0.9 | 0 | Marine13 | 130 ±90 | 80 ±180 |
| T1 | 18.5 | | | | | | 1000±440 | 265 ±335 |
| T2 | 47 | | | | | | 2250±520 | 1665±535 |
| Poz-112951 | 74 ±1 | Pk foraminifera | 2925±35 | 0.7 | 0 | Marine13 | 2710±85 | 2720±170 |
| T3 | 85.5 | | | | | | 4030±150 | 3785±500 |
| T5 | 172 | | | | | | 7385±110 | 7140± 27 |
| Poz-112952 | 189 ±1 | Pk foraminifera | 6860±50 | >1 | 150±30 | Marine13 | 7240±130 | 737 ±170 |
| Poz-113951 | 473 ±1 | Pk + Bk foraminifera | 13910±180 | 0.04 | Too little carbon retrieved | | | |
| T6 | 268.5 | | | | | | 7700±115 | 7810±195 |
| **P362/2-99 (P99)** | | | | | | | | |
| Poz-112947 | 11 ± 1 | Pk foraminifera | 415 ±30 | 0.8 | 0 | Post-bomb curve NHZ2 | 80 ±15 | 40 ±135 |
| T1 | 31.5 | | | | | | 1000±440 | 1150±615 |
| T2 | 54.5 | | | | | | 2250±520 | 2590±800 |
| T3 | 73.5 | | | | | | 4030±150 | 4255±920 |
| Poz-112948 | 79 ±1 | Pk foraminifera | 5990±50 | >1 | 0 | Marine13 | 6400±120 | 5250±960 |
| T4 | 91 | | | | | | 5965±140 | 6155±620 |
| T5 | 112 | | | | | | 7385±110 | 7170±285 |
| Poz-112949 | 322 ±2 | Pk foraminifera | 7890±60 | 0.5 | 150±30 | Marine13 | 8210±150 | 8200±190 |
| Poz-112950 | 518 ±1 | Pteropod | 8390±60 | >1 | 150±30 | Marine13 | 8770±200 | 9070±275 |

138

### 3.2 Age determination and transformation depth-to-age

The chronology of core P33 has been constructed based on a set of 14 radiocarbon samples (Blanchet et al., 2013). We prepared another 8 samples for radiocarbon dating using planktonic foraminifera and pteropod shells, which provide tie-points for cores P73 and P99 (Table 2). Only one sample in core P73 (Poz-113951) did not generate enough carbon for an accurate dating. The new radiocarbon measurements were performed at the Poznan radiocarbon laboratory (Poland).

More detailed stratigraphic constrains were obtained from correlation of the titanium over calcium (Ti/Ca) records
of cores P73 and P99 with core P33 (Fig. 2). It was shown that changes in sedimentation rates are coherent on the
western Nile DSF (Hennekam et al., 2015) and Ti/Ca records show similar patterns (Fig. S1). Six tie-points (marked
T1-T6 on Fig. 2) mark changes in Ti/Ca records identified in the upper parts of the cores and were used to further
synchronise the records of P73 and P99 with core P33 (Table 2).
The age-depth modelling was performed using Bacon version 2.3 (Blaauw and Christen, 2011), which enables to
discriminate between sections of the core with contrasting accumulation rates and to provide these as priors.
Convergence and mixing of the Markov chain Monte Carlo iterations used to build the age model by Bacon was
tested and the number of iterations was adjusted to obtain a Gelman and Rubin Reduction factor below 1.05
(Blaauw and Christen, 2011; Brooks and Gelman, 1998). The new age-depth model of core P33 based on
radiocarbon ages is similar to that presented in Blanchet et al. (2013), but age uncertainties are now available for
the whole core (Table 2 and Fig. 2a). Radiocarbon ages and the six tie-points were used for age modelling for cores
P73 and P99 using the Bacon program following a similar procedure to that used for core P33 (Fig. 2b, c) (Blaauw
and Christen, 2011). For the lowest part of core P73, for which there is no tie-point, we evaluated the ages and
uncertainties using the range of sedimentation rates observed in the other two cores for ages older than 7500 years
and derived the median sedimentation rate applying the maximum and minimum sedimentation rates observed as
uncertainty range (Fig. 2b). Age uncertainties range from 100 to 1000 yrs in core P33 (larger in the last 2000 yrs),
from 100 to 700 yrs in P73 (larger in the last 6000 yrs) and 10 to 1000 kyr in P99 (larger in the last 7000 yrs) (Fig.
163 2).

**3.3 Microfacies description and scanning electron microscopy**
A detailed examination of the microfacies was performed on the laminated part of core P33 (i.e., between 100 and
559 cm) and on a few samples of cores P73 and P99. Sediment blocks of 10 cm-long, 2 cm-wide and 1 cm-thick
were cut out of the fresh sediment with 2-cm overlaps to enable continuous microfacies analysis. Preparation of
thin-sections from soft and wet sediment blocks followed a standard procedure minimizing process-induced
disturbances of sediment micro-structures and included shock-freezing with liquid nitrogen, freeze-drying for 48
h, and epoxy resin impregnation under vacuum (Brauer and Casanova, 2001).
Detailed microfacies analysis was performed on the overlapping series of large-scale petrographic thin sections.
Microscopic analysis included the investigation of sediment using a petrographic microscope with non-polarized
and (cross)-polarized light, at 5x–40x magnifications (Carl Zeiss Axioplan) (Fig.3a). Backscatter scanning electron
microscope (SEM) images and energy dispersive spectroscopy (EDS) elemental analyses were acquired at GFZ
Potsdam using a Phenom™ XL (Fig. 3b). SEM and EDS analyses were performed on dried power samples mounted
on a graphite coated sample holder.

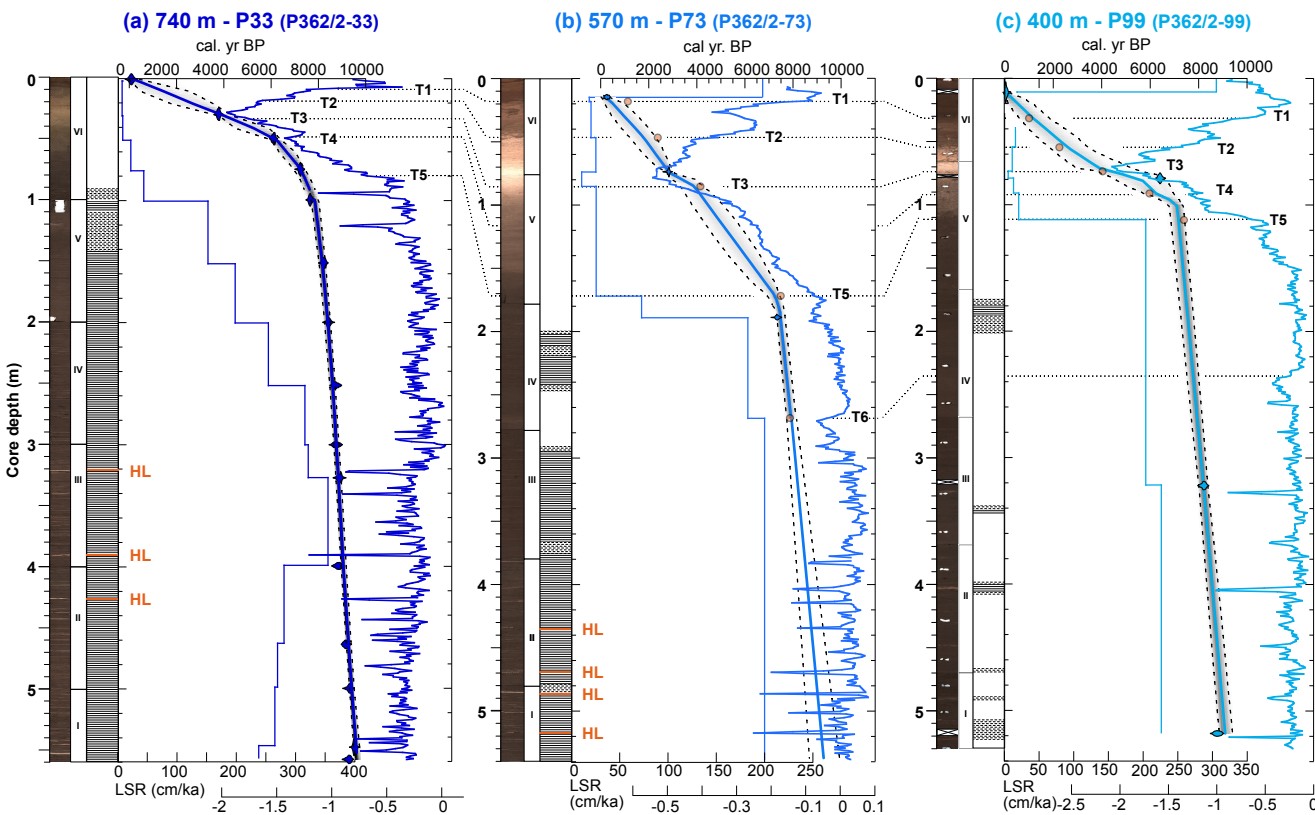

**Figure 2. Age models. (a) core P33, (b) core P73 and (c) core P99. For all cores, from left to right: core depth (m), half-core surface**
**images (for core P99, realised after sampling with empty sections shown as crosses), section number (roman numbers), schematic**
**lithological log (white: bioturbated, dashed: faint laminations, hatched: laminations, orange: hard layers labelled HL), linear**
**sedimentation rates (cm/ka, step curves), age-depth relationships as computed by Bacon V2.3 (Blaauw and Christen, 2011) with**
**radiocarbon ages (blue violin distributions) and tie points (orange dots), Log(Ti/Ca) records with tie-points T1-T6 (dashed lines)**
**(Tab. 2).**
**3.4 Mineralogy**
The mineralogical composition was determined by X-ray diffractometry (XRD) on random powder samples of core
P33 (Fig. 3c). Therefore, the rock chips were powdered to a grain size of < 63 μm, and loaded from the back side
of the sample holders. XRD analyses were performed with a PANalytical Empyrean x-ray diffractometer operating
with Bragg-Brentano geometry at 40 mA and 40 kV with CuKα radiation and PIXel3D detector at a step size of
0.013 °2θ from 4.6 to 85 °2θ and 60 sec per step. The Mineralogy was determined qualitatively with the EVA
software (version 11.0.0.3) by Bruker.

## 3.5 X-Ray Fluorescence scanning

The bulk sediment compositions for cores P33 and P99 were measured using an Avaatech[TM] X-Ray fluorescence (XRF) core-scanner at the Institute of Geosciences of the University of Kiel (Germany) (Table S1 and S2). Non-destructive XRF core-scanning measurements were performed every 1 cm using a Rh X-Ray source at 10 kV and 0.65 mA to acquire the elements S, Cl, K, Ca and Ti. Core P73 was measured using an ITRAX XRF core-scanner at GFZ Potsdam (Germany) (Table S3). These measurements were conducted every 1 cm with a Cr- X-Ray source operated at 30 kV and 60 mA to cover the same elements as acquired for core P33 and P99. Element intensity records obtained by XRF core-scanning are susceptible to down-core variations of physical properties, sample geometry and non-linear matrix effects (Tjallingii et al., 2007). These effects can be minimized by log-ratio transformation of element intensities, which also provide the most easily interpretable signals of relative changes in chemical composition (Weltje et al., 2015; Weltje and Tjallingii, 2008).

Two epoxy embedded samples of core P33 that are representative for the fine (P33-5 77-87 cm) and thick (P33-2 59-69 cm) laminations have been selected for $\mu$-XRF element mapping (Fig. 4). Measurements are conducted every 50 $\mu$m at 50 kV, 600 $\mu$A and 50 ms using a Bruker M4 Tornado, which is equipped with a Rh X-ray source in combination with poly-capillary X-ray optics generating an irradiation spot of 20 $\mu$m. Mapping results of the element K, Ca, and Ti representing solid-state chemical components are presented as normalized element intensities after initial spectrum deconvolution (Fig. 4). However, elements that occur predominantly in pore fluids (e.g. Cl and S) are not well preserved in epoxy embedded samples.

## 3.6 Stable oxygen and carbon isotopes

Measurements of stable oxygen and carbon isotopes on core P33 were realized both at GEOMAR and GFZ (Table S4). Carbonate-rich layers were sampled individually and all samples were subsequently freeze-dried and carefully ground. In order to dissolve only the carbonates in the powdered samples, we used orthophosphoric acid at 70 ℃ in a Kiel II (GEOMAR) or Kiel IV (GFZ-Potsdam) carbonate device. The $CO_2$ released was analysed using a Thermo Scientific MAT 252 (GEOMAR) or MAT 253 (GFZ-Potsdam) isotope ratio mass spectrometer (IRMS). The oxygen and carbon isotopic composition were calibrated against international limestone standard (NBS 19) and are reported relative to the Vienna Peedee Belemnite (VPBD) in the delta notation. Analytical precision of 0.06 ‰ for both the $\delta^{18}O$ and $\delta^{13}C$ measurements were determined from repeated analyses of internal limestone standards.

## 3.7 Biomarkers

All biomarkers (tetraether lipids, *n*-alkanes and alkenones) were measured at the organic geochemistry laboratory of NIOZ (Texel, Netherlands) on cores P33 and P99 (Table S5 and S6). The lipids were extracted from 63 sediment samples with a DIONEX Accelerated Solvent Extractor 200 at the NIOZ using a solvent mixture of 9:1 (v/v) dichloromethane (DCM)/methanol (MeOH). After addition of internal standards $C_{22}$ anti-isoalkane (*n*-alkanes), 10-nonadecanone (alkenones) and $C_{46}$ glycerol trialkyl glycerol tetraether, the total lipid extract was separated into apolar, ketone and polar fractions using pipette column chromatography loaded with aluminium oxide and the solvent mixtures 9:1 (v/v) hexane/DCM, 1:1 hexane/DCM and 1:1 DCM/MeOH as eluents, respectively. The apolar fraction was then separated into saturated hydrocarbon (long-chain odd *n*-alkanes and the $C_{22}$ anti-iso standard) and aromatic fractions using pipette columns loaded with $Ag^+$-impregnated silica and hexane and ethylacetate as eluents, respectively.

Molecular identification of the alkenones and *n*-alkanes was performed on a Thermo Finnigan Trace Gas Chromatograph (GC) Ultra coupled to a Thermo Finnigan DSQ mass spectrometer (MS). The alkenones and the *n*-alkanes were quantified with gas chromatography coupled to flame ionization detection. For GC analysis we used an Agilent HP 6890 GCs with CP Sil-5 columns (50 m length for the alkenones and 25 m for the *n*-alkanes) and helium as the carrier gas. Within the saturated hydrocarbon fraction, we also identified lycopane (2,6,10,14,19,23,27,31-octamethyldotriacontane) and several pentacyclic triterpenes (molecular mass M=440, thereafter referred to as "triterpenoids") such as $\beta$-amyrin Methyl Ether (ME) (Olean-12-ene-3$\beta$-ol), Germanicol ME (Olean-18-ene-3$\beta$-ol) and Taraxerol ME (Taraxer-14-en-3$\beta$-ol) (Fig. S2).

The polar fraction of 61 samples, containing the glycerol dialkyl glycerol tetraethers (GDGTs), was dissolved in a mixture of 99:1 (v/v) hexane/propanol and filtered through 0.45 mm PFTE filters. GDGTs were analysed (in triplicate) by high performance liquid chromatography (HPLC)/MS in single ion monitoring mode on an Agilent 1100 series LC/MSD SL (Hopmans et al., 2004; Schouten et al., 2007).

## 4 Results

### 4.1 Sedimentary patterns on the western Nile deep-sea fan

The deeper core P33 exhibits a near-continuous 500-cm finaly laminated interval between ca. 100 cm core depth and the bottom of the core, with high Ti/Ca ratios and high sedimentation rates (between 100 and 350 cm/ka) (Fig. 2a). A 40-cm interruption with poorly defined laminations and traces of bioturbation occurs in the upper part of the laminated interval between 140 and 100 cm core depth. The intermediate depth core P73 shows a series of laminated

intervals between 200 cm and the bottom of the core, which are accompanied by high Ti/Ca ratios and high sedimentation rates (between 150 and 250 cm/ka) (Fig. 2b). Several interruptions with poorly-laminated to non-laminated (bioturbated) intervals are identified at 480 cm, 370 cm, 240-290 cm and 215 cm. In contrast to the two deeper cores, the shallower core P99 is generally non-laminated with the exception of faint to well-defined laminations at 510-520 cm, 490 cm, 470 cm, 405 cm, 340 cm, and 180-200 cm core depth (Fig. 2c). However, both the Ti/Ca records and sedimentation rates exhibit changes similar to the two deeper cores with higher values between 200 cm core depth and the bottom of the core (sedimentation rates of ca. 200 cm/ka).

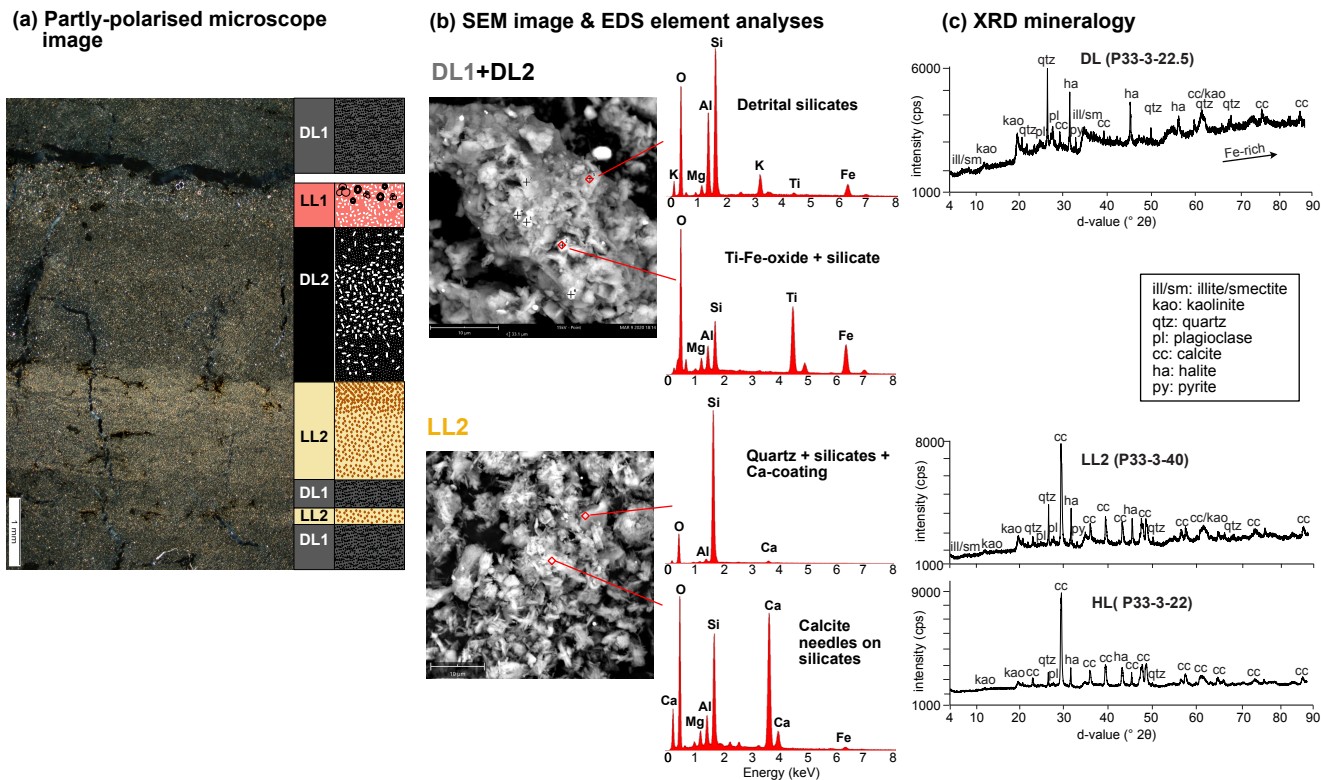

**Figure 3. Microscopic, elemental and mineralogical characterisation of sublayers from core P33. (a) Micro-facies and mineralogy of a typical sublayer succession in cores P33. Left to right: part-polarized microscope image and schematic log of sublayers succession DL1-LL2-DL2-LL1. (b) SEM images of powdered dark (DL1+DL2) and light sublayers (LL2) with EDS elemental identification. (c) mineralogical assemblage of powdered DL, LL and HL samples using XRD scans (from 4 to 85 °2θ). In sample DL (P33-3-22.5), the increase of the intensity baseline towards higher angles is due to the presence of an Fe-rich phases, probably pyrite.**

## 4.2 Structure and composition of the laminations

The lamination structure is regular throughout core P33 and similar to the laminations in core P73 and P99 (Fig. S4). Well-defined laminations are composed of an alternation of dark and light bands forming sublayers, whereby dark sublayers (DL) are mostly thicker (0.2 and 15 mm) than light sublayers (0.1 to 0.4 mm) (Fig. 3a). The

mineralogy of DL consists of clay minerals (kaolinite, illite and smectite) as well as silt-size detrital minerals (quartz, plagioclase, pyroxene, iron-titanium oxides) (Fig. 3c). Subdivisions of DL into potassium (K)-rich (labelled DL1) and titanium (Ti)-rich (labelled DL2) parts are observed in thicker laminations. These can be distinguished by XRF elemental mapping as well as by their texture, Ti-rich sublayers containing coarser particles (ca. 20-30 μm) as K-rich sublayers (Fig. 3a, 4b).

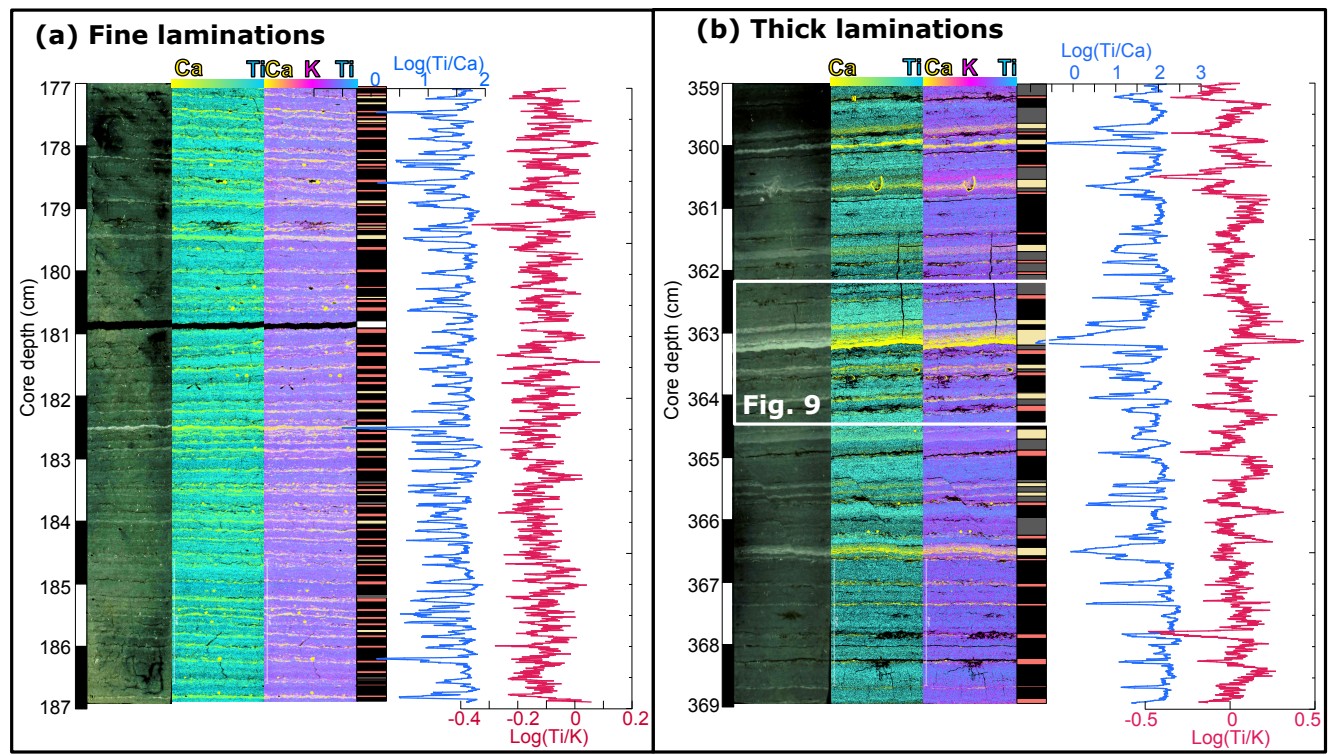

**Figure 4. Laminations in core P33. (a) Fine laminations and (b) thick laminations from sections located at ca. 180 and 360 cm core depth, respectively. From left to right: core depth (cm), cross-polarized scan of the thin section, elemental mapping showing relative amounts of calcium (Ca, yellow) versus titanium (Ti, turquoise), elemental mapping showing relative amounts of calcium (Ca, yellow), potassium (K, pink) and titanium (Ti, blue), schematic lithological log with color-coded sublayer type following Fig. 3a (LL1: pink, DL1, grey, LL2: yellow and DL2: black), down-section profile of log(Ti/Ca) (blue) and log(Ti/K) (red).**

Light sublayers (LL) contain predominantly carbonate and can be subdivided into two main types (Fig. 3a). The light-sublayer type 1 (LL1) are diffuse sublayers of silt-sized (ca. 10-20 μm) and unsorted particles containing biogenic carbonates (coccoliths and foraminifera), organic matter and quartz grains (Fig. 3a). Foraminifera shells generally contain small grains of iron sulphides, which may have formed on organic residues (Fig. S5). Some of these sublayers present a slight cementation. The light sublayers type 2 (LL2) contain well-sorted fine (ca. 1-10 μm) and needle-shaped calcite minerals, mixed with detrital silicate grains, including clay minerals (Fig. 3b). The

detrital assemblage in LL2 is similar to that of dark layers but contains a larger amount of calcite (Fig.3c). LL2 are
abiotic, have a wavy internal structure and contain concentrated lenses of carbonate grains (Fig. 3a). Some of these
sublayers show a sharp lower boundary and either an upward decrease or increase in grain concentration. In the
lower part of core P33, three prominent LL2 are cemented and consolidated (thereafter denoted "hard layers" HL)
and are generally thicker than the softer LL2 (Fig. 2a). Four similar HL are also found in core P73 (Fig. 2b). The
presence of organic matter is observed either within LL1 or at the base of DL1.

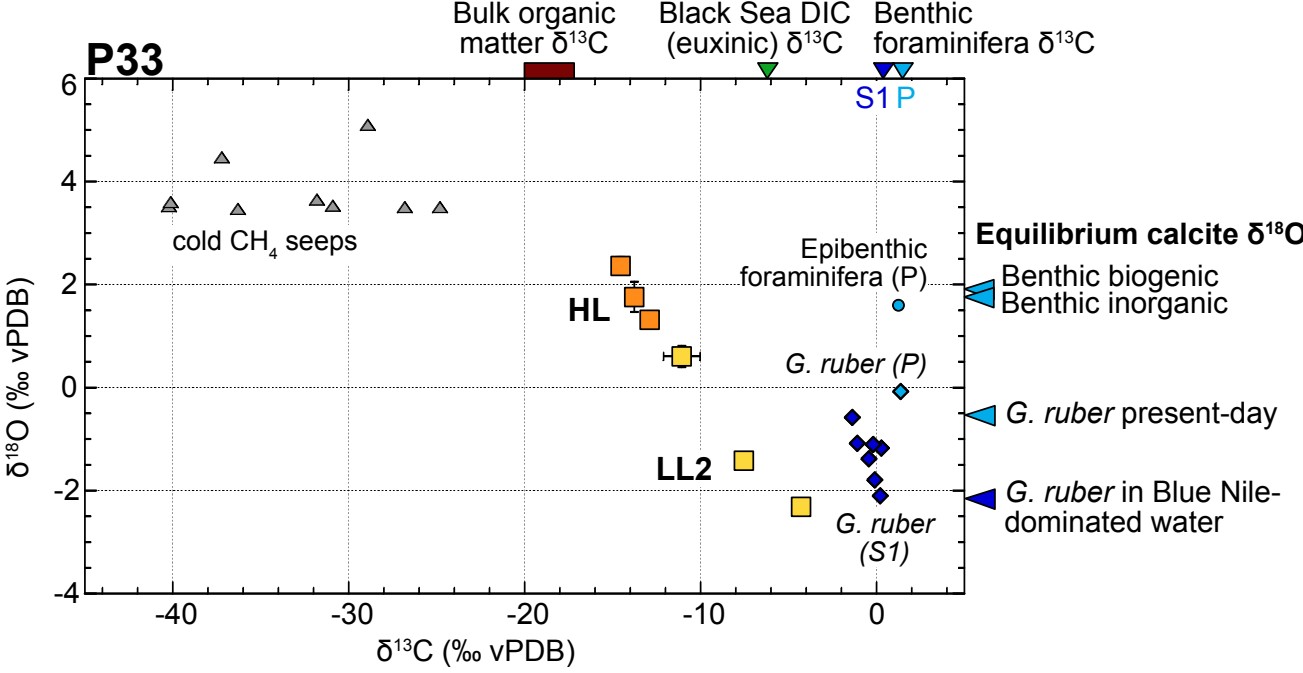

**Figure 5. Isotopic signature of LL2 and HL from core P33. Scatter plot of δ¹⁸O vs δ¹³C values of three LL2 (yellow) and three**
**cemented HL (orange), as compared to isotopic signatures of methane-derived carbonates from the Nile DSF (grey triangles) (Aloisi**
**et al., 2000; Bayon et al., 2013); P33 top-core and S1 planktonic foraminifera (light and dark blue diamonds, resp.); present and S1**
**epibenthic foraminifera (light blue circle and blue triangles; Grimm et al., 2015); P33 bulk organic matter δ¹³C (brown rectangle);**
**Black Sea δ¹³C of dissolved inorganic carbon (DIC) (green triangle) (Fry et al., 1991); δ¹⁸O value of biogenic and inorganic benthic**
**and planktonic calcite formed in equilibrium with seawater (blue triangles) (Table S4, see calculation details in Supplementary**
**Information).**
The sublayers are deposited in a remarkably constant succession over the length of the laminated interval (Fig. 4).
This succession is particularly clear in the thicker laminations and follows the pattern from bottom to top: LL1,
DL1, LL2 and DL2 (Fig. 4b). Sublayers LL1 and DL2 are always present in the succession, whereas DL1 is very
thin or not detectable in ca. 10% of the successions and LL2 is missing in ca. 30% of the successions. Sublayers
are on average thicker in the lower part of the of the laminated interval (between 260 and 570 cm) and thinner in
the upper part. At our core sites, we did not identify any remains of diatoms. Thirteen anomalously thick layers
(mm- to cm-scale) were identified throughout the record and labelled event layers (EL). These well-mixed layers
containing matrix-supported grains and showing a clear bimodal distribution of silt- (ca. 15-20 µm) and mud-sized
particles. In general, EL have sharp but non-erosive boundaries and either no clear internal structure or show a
fining upward.

**Table 3: Overview of the proxies used and their main interpretation**

| Proxy | Abbreviation | Interpretation | References |
|---|---|---|---|
| **Oxygenation** | | | |
| Laminations | | Absence of bioturbation due to the lack of oxygen in bottom waters, which prevents burrowing animals to live in the sediments, changes in seasonality | (Schimmelmann et al., 2016) |
| Log(S/Cl) ratio | S/Cl | Relative increase of sulphur with respect to pore-water concentrations indicating S accumulation due to the formation of iron sulphides, which form when deoxygenation increases. | (Liu et al., 2012; Revel et al., 2015) |
| Lycopane | | Low oxygen levels | (Sinninghe Damsté et al., 2003) |
| Benthic foraminifera | BF | Existence/absence of oxygen in the bottom waters | (Abu-Zied et al., 2008; Le Houedec et al., 2020) |
| **Primary productivity** | | | |
| Alkenones | | Abundance of the haptophyte algae coccolithophorid *Emiliana Huxley* in the surface waters | (Marlowe et al., 1984; Volkman, 2000) |
| Crenarchaeol | | Abundance of thaumarcheota living in the upper part of the water column | (Besseling et al., 2019) |
| **Sedimentation and river runoff** | | | |
| Log(Ti/Ca) ratio | Ti/Ca | Relative variation of terrigenous versus marine sediments input | (Billi and el Badri Ali, 2010; Blanchet et al., 2013; Garzanti et al., 2015) |
| Log(Ti/K) ratio | Ti/K | Relative proportion of coarser Ti-rich minerals to finer K-rich clay minerals | (Billi and el Badri Ali, 2010) |
| Sedimentation rate | SR | Speed of sediment accumulation (in cm/ka) | |
| Branched GDGTs | brGDGTs | Soil organic matter (with potential minor contributions from riverine and marine organic matter) brought by river runoff | (De Jonge et al., 2014; Sinninghe Damsté, 2016; Xiao et al., 2016) |
| Triterpenoids and long-chain n-alkanes | Triterpenoids and n-alkanes | Terrestrial higher plants biomarkers brought by river runoff | (Blanchet et al., 2014; Bray and Evans, 1961; Diefendorf et al., 2012; Jacob et al., 2005) |
| **Water-mass physical & chemical properties** | | | |
| Oxygen isotopes | $\delta^{18}O$ | Changes in temperature, salinity and isotopic composition (often related to the origin) of the water | |
| Carbon isotopes | $\delta^{13}C$ | Changes in organic matter consumption and biogeochemical processes in the water column | |


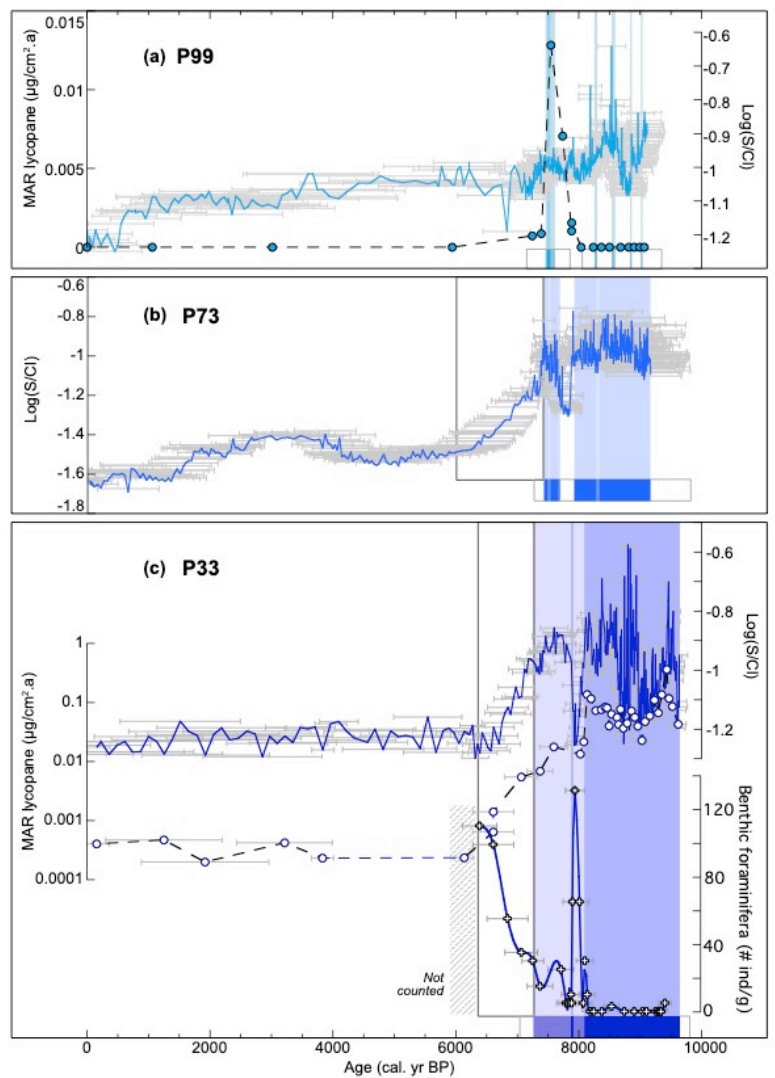


**Figure 6. Proxies of paleo-oxygenation in cores (a) P99 (turquoise), (b) P73 (blue) and (c) P33 (dark blue) on age scale (0-10 ka BP).**
**For each core, from top to bottom: Log-ratio of S/Cl (continuous lines), mass accumulation rates (MAR) of lycopane in P99 and P33**
**(dashed lines and circles), lamination pattern (darker boxes: laminated, lighter boxes: faint laminations). For P33: bottom curve**
**shows the number of benthic foraminifera per g of sediment (white crosses and cubic spline as thick blue curve). Open boxes in P73**
**and P33: interval of gradual reoxygenation. Grey bars under each marker and around laminated intervals show the age uncertainty.**

Stable $\delta^{18}O$ and $\delta^{13}C$ isotope measurements were performed on a set of 6 LL2 and HL sublayers ranging in
thickness from 0.1 to 0.8 mm (Fig. 5). Cemented HL have a strongly depleted $\delta^{13}C$ (-10 to -15 ‰) and an enriched
$\delta^{18}O$ (0.5 to 2.5 ‰) signature compared to the softer LL2 (with $\delta^{13}C$ between -11 and -5 ‰ and $\delta^{18}O$ between 0.5
and -2 ‰). These carbonates show a wide range of stable oxygen and carbon isotope values and apparently fall on
a mixing line between a depleted $\delta^{13}C$/heavy $\delta^{18}O$ and an enriched $\delta^{13}C$/light $\delta^{18}O$ end-member, which do not
correspond to either biogenic carbonates or regional methanogenic carbonates (Fig. 5).

## 4.3 Spatial and temporal variations

To build on the results from sedimentary facies at mm-scale, we explored the variations of oxygenation, primary
productivity and terrigenous input during the past 10 kyr over the Nile DSF. We used a wide range of proxies,
which are listed in Table 3 together with their interpretation keys.
In order to reconstruct past changes in oxygenation conditions, we used four independent tracers: the presence of
laminations and lycopane, the benthic foraminifera assemblage and the sulphur to chlorine (S/Cl) ratio (Table 3
and Fig. 6). In all cores, laminated intervals coincide with high sedimentation rates (> 100 cm/ka, Fig. 6).
In the shallowest core P99, several short-term faintly-laminated intervals are observed between 9 and 8.3 ka BP
and between 7.6 and 7.4 ka BP, which contain well-preserved laminations and a peak value of lycopane mass
accumulation rates (MAR) (up to 0.013 $\mu g/cm^2$.a) at 7.5 ka BP (Fig. 6a). High S/Cl values occur between 9 and
8.5 ka BP, but are interrupted by a strong drop around 8.7 ka BP. Values decrease after 8.5 ka BP, but remain
slightly elevated between 8.5 and 7.2 ka BP (Fig. 6a). In intermediate depth core P73, laminations occur as a nearly
continuous interval between 9.2 and 7.4 ka BP with an interruption between 8.3 and at 7.6 ka BP (Fig. 6b). High
S/Cl values occur between 9.6 and 7.5 ka BP, with a strong drop in the non-laminated interval around 7.9 ka BP.
The S/Cl record shows a gradual decrease between 7.3 and 6 ka BP, after which the record remained low and only
varied slightly (Fig. 6b). In the deepest core P33, laminations occur continuously between 9.6 and 8.1 ka BP,
followed by an interval of faint laminations until 7.3 ka BP (Fig. 6c). Lycopane MAR follow the lamination pattern
with high values in the laminated interval between 9.6 and 8.1 ka BP (between 0.1 and 0.35 $\mu g/cm^2$.a), when MAR
abruptly drop to ca. 0.01 $\mu g/cm^2$.a and then gradually decrease until ca. 6 ka BP to less than 0.001 $\mu g/cm^2$.a) (Fig.
6c). The S/Cl record shows generally higher values between 10 and 7.5 ka BP with lower values between 9.5 and
9.0 and around 8.8, 8.3, and 8 ka BP (Fig. 6c). A steady decrease is observed between 7.5 and 6.5 ka BP followed
by low values in the younger part of the core. The laminated interval between 9.6 and 8.1 ka BP is devoid of benthic
foraminifera (except at ca. 9.5 and 8.55 ka BP). From then on, a few occurrences are observed with a peak around
7.9 ka BP but their presence remains very volatile until ca. 7.4 ka BP, after which their number grows gradually
(Fig. 6c).

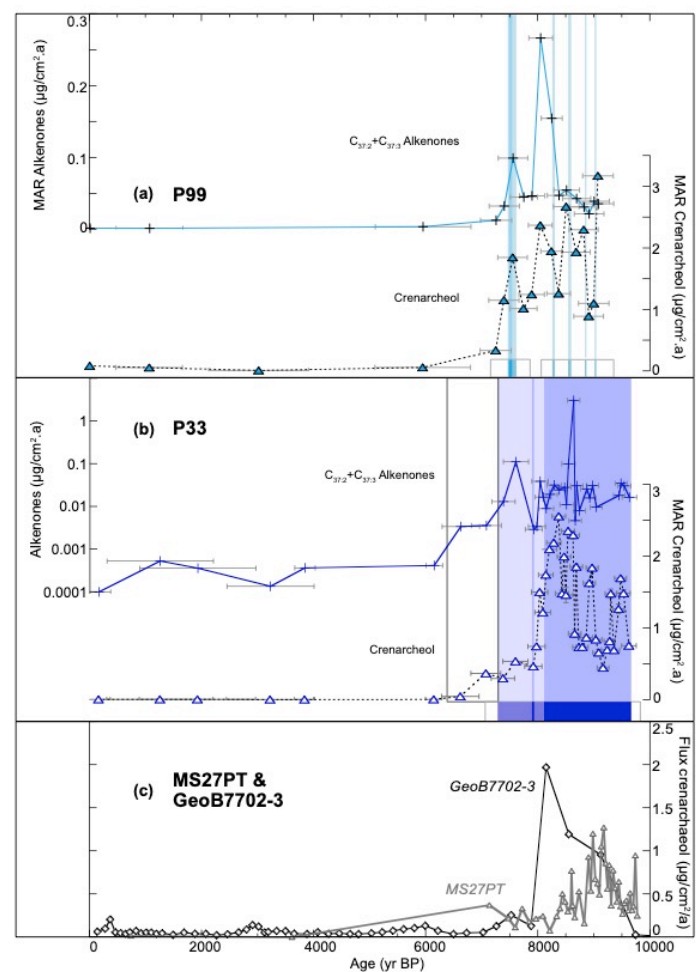


**Figure 7. Proxies of primary productivity in cores (a) P99 (turquoise), (b) P33 (blue), (c) GeoB7702-3 (black) and MS27PT (grey) on**
**age scale (0-10 ka BP). MAR of alkenones (crosses and continuous lines) and MAR of crenarchaeol for cores P33 and P99 (triangles**
**and dashed lines), as well as for cores GeoB7702-3 (black diamonds) (Castañeda et al., 2010) and MS27PT (grey triangles) (Ménot**
**et al., 2020). Lamination pattern (similar colour coding as in Fig. 6). Grey bars show the age uncertainty.**

Changes in primary productivity off the Nile mouth was traced using MAR of $C_{37}$-alkenones and crenarchaeols in
cores P99 and P33 (Table 3 and Fig. 7). Alkenones and crenarchaeol MAR are of similar order of magnitude in
both cores (up to 0.3 $\mu$g.cm$^{-2}$.a$^{-1}$ and 3.5 $\mu$g.cm$^{-2}$.a$^{-1}$, respectively). The overall patterns of changes in biomarkers
are similar in both cores, with highest fluxes being measured between 10 and 7 ka BP. Several short pulses of
alkenone and crenarchaeol deposition are observed in shallowest core P99 at 9.1, 8.8, 8.5, 8 and 7.5 ka BP, although
the alkenone MAR do not show large peaks between 9 and 8.5 ka BP, while crenarcheol MAR are high (Fig. 7a).
There is a clear correspondence between the occurrence of peaks in MAR of organic compounds and the presence

of laminations (Fig. 7a). In deeper core P33, the sampling resolution is higher and the signal is spikier (Fig. 7b). The MAR of both crenarchaeol and alkenones are higher in the laminated interval between 9.6 and 8.2 ka BP and in the faintly laminated interval between 8.2 and 6.5 ka BP (Fig. 7b). In nearby cores MS27PT (1390 m) and GeoB7702-3 (562 m) (Fig. 7c), crenarchaeol MAR are also higher between 10 and 8 ka BP, but MAR are highest between 10 and 8.5 ka BP in core MS27PT while a marked peak occurred in core GeoB7702-3 at ca. 8.1 ka BP (Castañeda et al., 2010; Ménot et al., 2020).

The changes in river runoff and sediment input to the Nile DSF were reconstructed using major elemental contents, MAR of branched GDGTs (brGDGTs), odd long-chain *n*-alkanes and triterpenoids and sedimentation rates (Table 3, Fig. 8 and Fig. S3). The Ti/Ca and Ti/K records of all three cores ratios show very similar patterns with relatively high amounts of Ti in the early Holocene (ca. 10-7.5 ka BP) and the late Holocene (1-0 ka BP). Especially in well-laminated cores P73 and P33, high-frequency fluctuations in the Ti/Ca records during the early Holocene are related to the presence of Ca-rich laminations (Fig. 4). Although the magnitude is not similar in all cores, a rapid decrease in Ti/Ca records is observed around 7.2 ka BP, followed by a more gradual decrease until ca. 3-4 ka BP (Fig. 8). The decrease in Ti/Ca at 7.2 ka BP is accompanied by a simultaneous decrease in Ti/K (Fig. 8). A stepwise increase in Ti/Ca and Ti/K is observed in all cores between 3-3.5 and 1 ka BP. Sedimentation rates (SR) were calculated linearly between dated points (Table 2) (Fig. 8). They are of the same order of magnitude and vary synchronously in the three cores with highest SR between ca. 10 and 7 ka BP (>100 cm/ka). An abrupt decrease is recorded in cores P99 and P73 around 7 ka BP (Fig. 8a, b) whereas SR decreased more gradually from 9 to 7 in core P33 (Fig. 8c). After 7 ka, the SR remained below 20 cm/ka in all three cores. The MAR of *n*-alkanes (and triterpenoids, Fig. S3) are a factor 10 higher in core P33 than in core P99 (up to 7 and 0.6 $\mu g.cm^{-2}.a^{-1}$, resp.) but brGDGTs MAR are similar in both cores (up to 4 $\mu g.cm^{-2}.a^{-1}$ and 2 $\mu g.cm^{-2}.a^{-1}$, resp.) (Fig. 8a, c). Higher fluxes of terrestrial organic matter are observed in shallowest core P99 between 9.5 and 7.5 ka BP with pulses at 9.1, 8.8, 8.5, 8.2 and 7.5 ka BP in laminated intervals (Fig. 8a). In deeper core P33, the MAR of terrestrial biomarkers are higher in the laminated interval between 9.6 and 8.2 ka BP and lower in the faintly laminated interval between 8.2 and 6.5 ka BP (Fig. 8c).

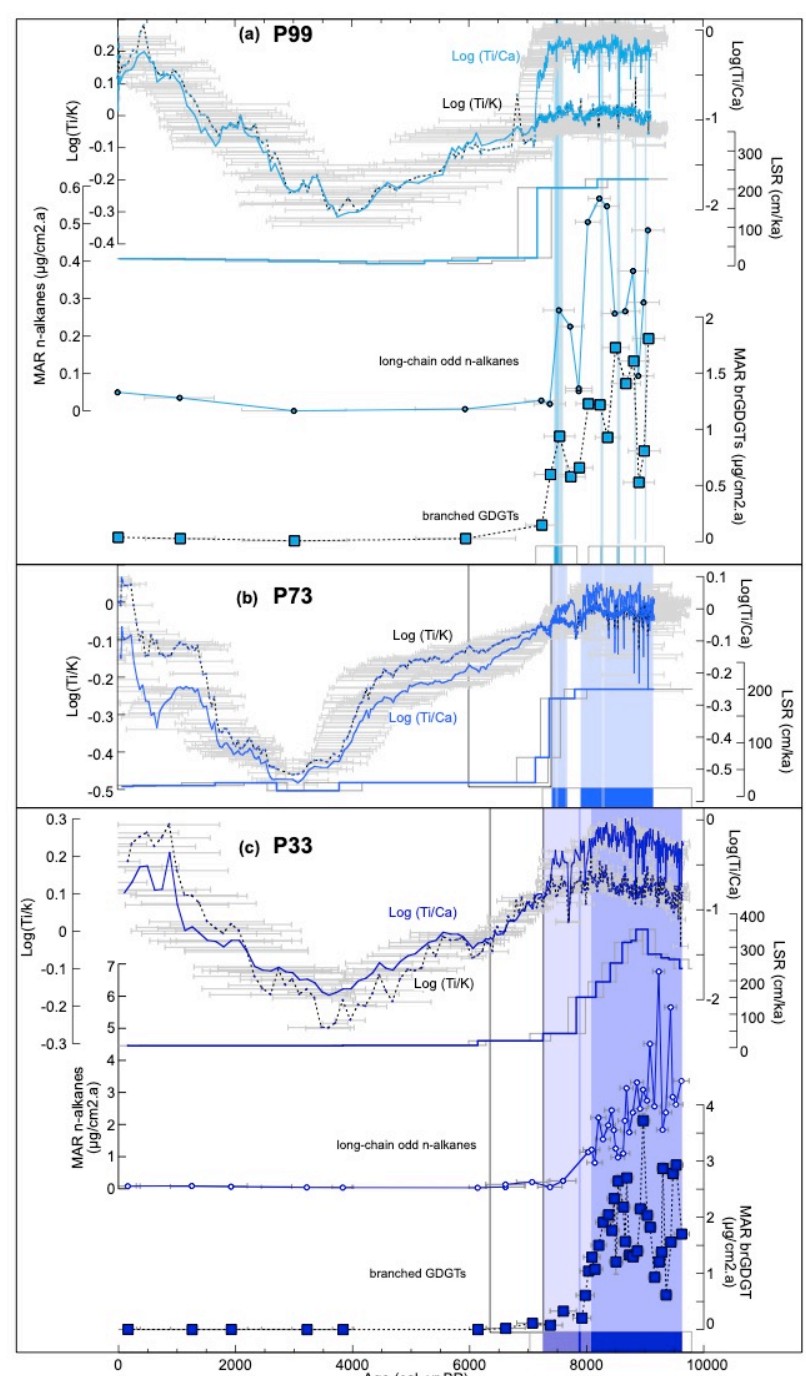

**Figure 8. Proxies of terrigenous input in cores (a) P99 (turquoise), (b) P73 (blue) and (c) P33 (dark blue) on age scale (0-10 ka BP).**
**For all cores: Log(Ti/Ca) (continuous lines) and Ti/K (dashed lines), sedimentation rates and lamination pattern (similar colour**
**coding as in Fig. 6). For cores P99 and P33 (a, c): MAR of odd long-chain *n*-alkanes (circles and continuous lines) and branched**
**GDGTs (squares and dashed lines). Grey bars show the age uncertainty.**

## 5. Discussion

### 5.1 Seasonal dynamics over the Nile deep-sea fan as derived from microfacies analyses

As shown in Fig. 5, the sequence of dark and light sublayers (DL2, LL1, DL1, LL2) is consistent throughout the laminated interval, even though DL1 and LL2 can be very thin and sometimes absent. We propose that the different sublayers recorded distinct depositional regimes throughout the year and represent an annual cycle.

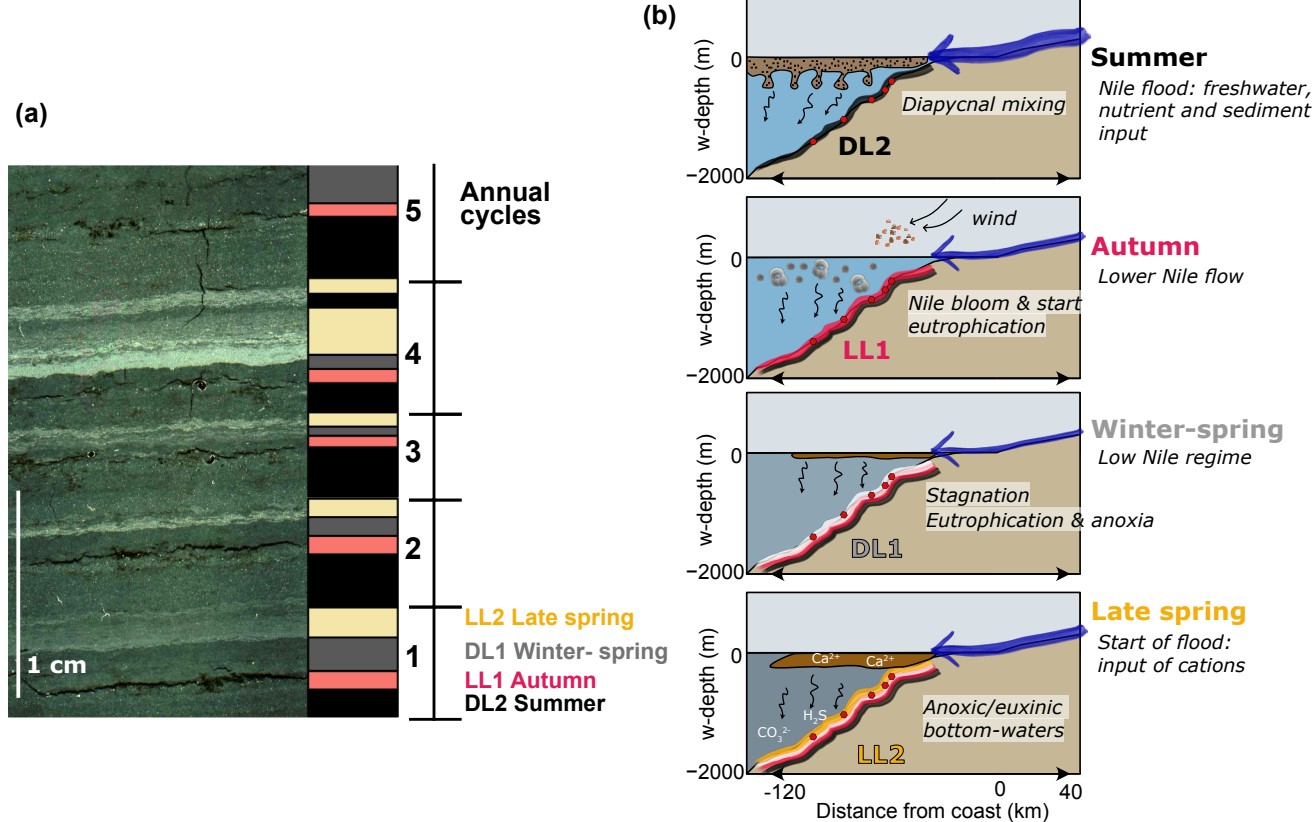

**Figure 9. (a) Seasonal succession of sublayers and (b) schematic seasonal cycle on the Nile DSF. (a) From left to right: part-polarized microscope image, sublayer identification, determined season for each sublayer type and number of annual cycles in the sample. (b) sketches of an annual cycle as derived from microfacies analyses from summer (top) to late spring (bottom).**

The coarser grained DL2 are proposed to represent sediments deposited during summer floods of the Nile River (Fig. 9). The presence of silt-sized particles of smectite, plagioclases and iron-titanium oxides in the DL2 matches the grain size of particulate matter transported during peak Nile discharge (Billi and el Badri Ali, 2010), as well as its mineralogical composition, typical for basaltic rocks of the Ethiopian volcanic plateau (Garzanti et al., 2015, 2018) (Fig. 3c). According to present-day and historical observations (Halim et al., 1967), summer floods lead to the formation of freshwater plumes above a saline seawater mass (Fig. 1a). During sapropel S1, this low-salinity

subsurface water mass extended much further offshore in the Levantine Basin (Bale et al., 2019; Tang and Stott, 1993; Vadsaria et al., 2019). The important sediment loading of the freshwater plume that entered the relatively saline Levantine Basin probably played an important role in driving sedimentation on the Nile DSF. Observations of mixing processes between freshwater plumes containing a high sediment load and the underlying saline seawater shows the formation of salt and sediment fingers below the plume that lead to a fast and efficient settling of sediments (Jazi and Wells, 2016; Parsons et al., 2001). On the Nile DSF, the deposition of flood particles is likely driven by the formation of sediment-laden salt fingers evolving into dense (hyperpycnal) sediment plumes (Fig 9b) (Ducassou et al., 2008). The thick DL2 observed throughout the record match the depositional expression of hyperpycnal plumes and associated Nile summer floods. In addition to this regular pattern, exceptional flood events have been identified in several sediment archives over the western deep-sea fan based on the occurrence of mud clasts, and correspond in our record to the thick EL (Mologni et al., 2020).

The detrital DL2 are overlain by coarse-grained light sublayers (LL1) that are enriched in organic matter and biogenic carbonates, mostly foraminifera (often *G. ruber*) and coccoliths (Fig. S5). High abundance of planktonic foraminifera and in particular of *G. ruber* during sapropel S1 was reported in a sediment core from the SE Levantine Basin also under the influence of the Nile outflow (Mojtahid et al., 2015). Although generally associated to oligotrophic conditions in the Mediterranean Sea (Pujol and Grazzini, 1995), it was found that *G. ruber* thrived in low salinity and nutrient rich waters of the Nile plume. This high planktonic productivity was related to sustained reproduction of foraminifera during the months following the Nile flood, when the salinity increased while food resources remained high (Mojtahid et al., 2015). We propose here that this process was similar to the so-called "Nile blooms", which occurred in September-October prior to the construction of the Aswan Dam in 1965 (Nixon, 2003). Fertilisation of the surface waters off the Rosetta mouth by large nutrients input during the summer Nile floods triggered an algal bloom, upon which zooplankton grazed (although no foraminifera was reported by Halim et al., 1967). We did not identify any diatom frustules associated to the Nile blooms, probably due to the poor preservation of opal in the Levantine Basin. However, the combination of foraminifera, coccoliths and organic matter in LL1 (Fig. S5) is proposed to represent post-flood layers following a fertilisation process similar to that occurring during the Nile blooms (Fig 9b). Efficient transport of nutrients from the Nile freshwater plume to the underlying saline Mediterranean water might have also been boosted by the process of diapycnal mixing described earlier (Oschlies et al., 2003). High levels of primary productivity probably triggered a gradual eutrophication of the underlying water masses due to bacterial oxidation of sinking organic matter. Occasionally, these layers contain quartz grains that might be related to autumn-early winter dust storms (Goudie and Middleton, 2001).

The following sublayer DL1 is generally a fine-grained homogenous dark layer, which is enriched in potassium and reflects autumn to spring deposition. Analysis of the present suspension load of the Nile River shows that the suspended load of the main Nile is enriched in clay particles during low Nile discharge periods (i.e., from late autumn to early summer) (Billi and el Badri Ali, 2010; Garzanti et al., 2006). At present, the core site is ventilated by formation of intermediate and deep-water masses as it lies at the transition between LIW and LDW flows (Kubin et al., 2019). During S1 deposition, it was suggested that a sluggish hypoxic water mass occupied the depth range 500-1800 m, originating from the Aegean Sea and labelled "Sapropel Intermediate Water" (SIW) (Zirks et al., 2019). The persistence of a yearly cycle of fluvial input during S1, with a marked low flow regime in winter as observed by our data fits previous observations by Tang and Scott (1993) and suggests that surface waters in the Aegean Sea might have been dense enough to sink and circulate at intermediate depth. However, the ventilation by SIW was probably very sluggish because we observe no reoxygenation of bottom waters during the winter or spring at our core site (see §5.2).

Sublayers DL1 are often overlain by a final sublayer of homogenous fine calcite grains (LL2) that is sometimes cemented (HL). The mineralogical assemblage in these layers is similar to the detrital-rich layers but shows an enrichment in calcite (Fig. 3c). SEM pictures suggest that microcrystalline calcite might have formed as a coating on the detrital grains, which could have served as nucleus (Fig. 3b). The internal draping structure of these layers and their regular occurrence in the lamination sequence always intercalated between DL1 and DL2 without sedimentary disturbance suggest that these calcite grains formed seasonally close to the seafloor, either in seawater or in highly porous sediments at the water-sediment interface (Fig. 9a). Furthermore, LL2 are also identified both in cores P73 and P99 with sublayer structures similar to those of core P33 (Fig. S4). This points to a regionally ubiquitous formation of authigenic carbonates in the water column, rather than local diagenetic processes occurring in the sediments. These layers are linked to the late spring-early summer season (Fig. 9b). The formation process of cemented HL is not clearly determined as these layers show internal disturbances and might have formed either at the seafloor (Allouc, 1990) or deeper in the sediments (Kasten et al., 2004).

## 5.2. Water-column stratification and chemistry

### 5.2.1. Bottom-water chemistry

The occurrence of bottom water-derived authigenic calcite sublayers (LL2) in core P33 provides a unique opportunity to investigate seawater chemistry using their oxygen and carbon isotope signatures (Fig. 5, 10). The $\delta^{13}C$ signatures of soft LL2 calcite sublayers range from -11 to -5 ‰ and are therefore distinct from $\delta^{13}C$ values of

present epibenthic foraminifera (ca. 1.2 ‰; Grimm et al., 2015) or from values estimated during S1 deposition (ca. 0.3‰; Grimm et al., 2015) (Fig. 5). The markedly depleted $\delta^{13}C$ of LL2 are comprised between the $\delta^{13}C$ of bulk organic matter in core P33 (-17 to -20 ‰) and bottom water dissolved inorganic carbon (DIC) in the euxinic Black Sea (-6.3 ‰, Fry et al., 1991) (Fig. 5, 10b). For the two older LL2 in core P33, the bathymetric $\delta^{13}C$ gradients from surface to bottom waters at the core site are similar to that of the present-day Black Sea: from 0 ‰ at the surface (as indicated by *G. ruber* $\delta^{13}C$) to ca. -7‰ in bottom waters offshore the Nile mouth (Fig. 10b), and from 0.8 in surface to -6.3 ‰ in bottom waters in the Black Sea (Fry et al., 1991). Based on these preliminary results, we propose that LL2 formed in euxinic (sulfidic) bottom waters dominated by respiratory processes (i.e., degradation of organic matter) but further analyses are required to confirm this hypothesis (e.g., isotopic signatures of molybdenum and iron; Matthews et al., 2017).

The $\delta^{13}C$ of HL and of the youngest LL2 is too low to result from anaerobic oxidation of organic matter only and suggests that part of the bottom-water DIC might have incorporated methane diffusing upwards from the sediments to bottom waters. As soft LL2 are assumed to form from bottom waters, the presence of methane implies a shoaling of the sulphate-methane transition zone (SMTZ) in the water column, which has been identified in the deeper parts of the Black Sea basin (Michaelis et al., 2002; Riedinger et al., 2010; Schouten et al., 2001). The persistence of methane in bottom waters of the SE Levantine Basin during sapropel S1 has also been proposed to account for the formation of Mg-rich calcite crusts around methane seeps and vents on deeper parts of the Nile DSF (Aloisi et al., 2002; Bayon et al., 2013). Anoxic and alkaline bottom waters at that time likely prevented the oxidation of methane into $CO_2$ and led to the precipitation of carbonate crusts with a very depleted $\delta^{13}C$ signature (-25 to -45 ‰), typical of hydrocarbon sources and clearly distinct from that of authigenic carbonate layers found in core P33 (Fig. 5). However, the incorporation of methane generated at the SMTZ from fermentation of sedimentary organic matter would provide a likely source of depleted DIC. It is therefore possible that methane-rich conditions developed temporarily in bottom waters and led to the formation of the youngest LL2 in core P33. The cemented HL in core P33 have a homogenous $\delta^{13}C$ value of ca. -15 ‰ and might either have formed at the seafloor from bottom water similarly to the youngest LL2, or from pore-waters deeper in the sediments during diagenetic precipitation of calcite around the SMTZ. Such processes have previously been attributed to non-steady state diagenesis and fixation of the SMTZ at a certain depth due to changes in sediment supply (Kasten et al., 2004).

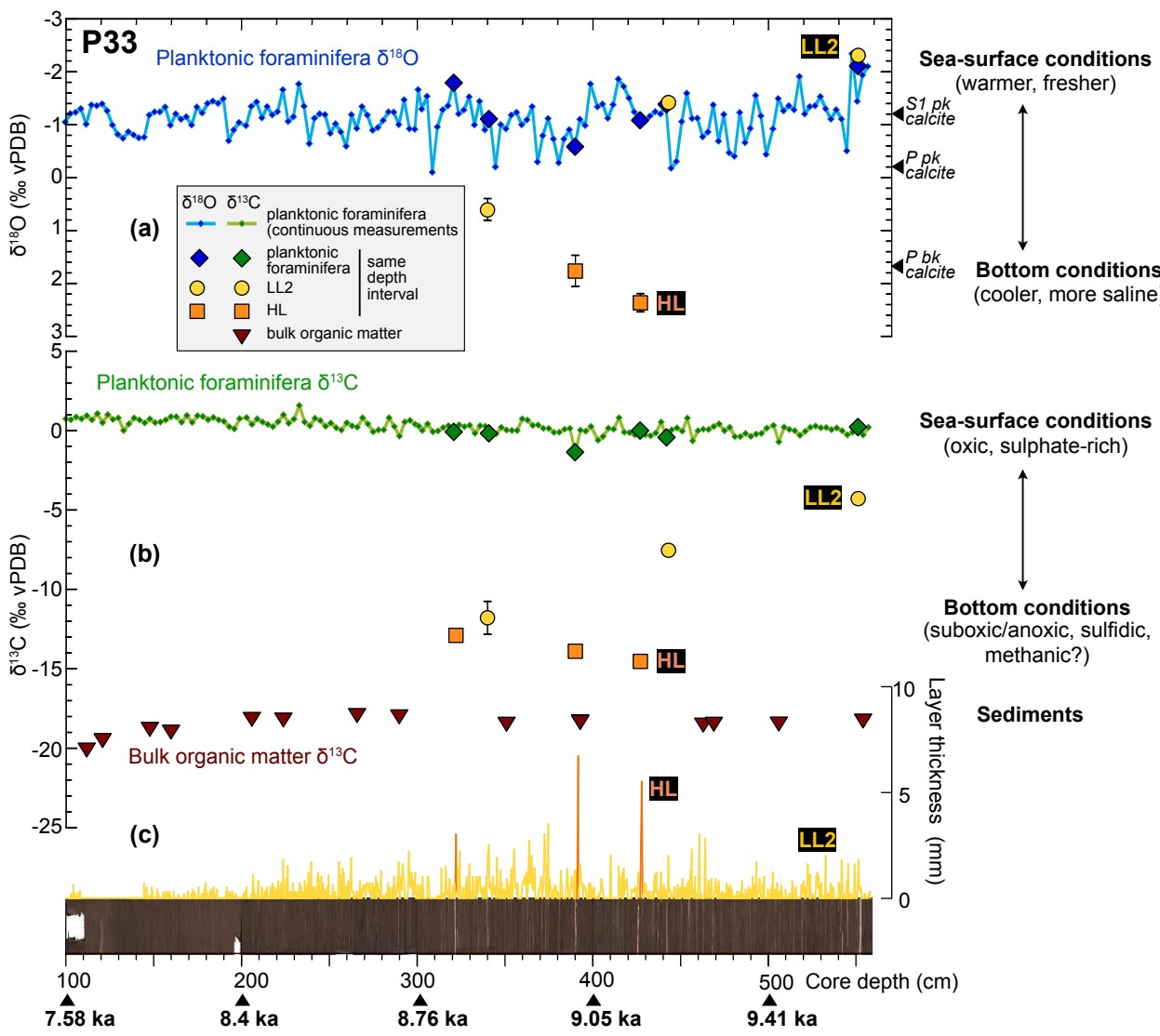

493

**Figure 10. Changes in water-column stratification and chemistry as provided by pilot measurements on authigenic calcite in individual LL2 and HL of core P33. (a) δ$^{18}$O of planktonic foraminifera (blue curve and diamonds), LL2 (yellow dots) and HL (orange squares), as compared to biogenic and inorganic calcite formed in equilibrium with seawater providing information on the water-column stratification as indicated on the right-hand side. (b) δ$^{13}$C of planktonic foraminifera (green curve and diamonds), LL2 (yellow dots) and HL (orange squares) and bulk organic matter (brown triangles) showing the changes in water-mass oxygenation state. (c) Thickness of LL2 and HL, picture of the core surface, depth scale and indications of age.**

500

### 5.2.2. Water-column stratification and precipitation of authigenic calcite

The $\delta^{18}O$ of LL2 and HL ranges from -2 to 2.5 ‰, which is comprised between the $\delta^{18}O$ signature of calcite formed in equilibrium with surface waters (ca. -1 to -2 ‰) and bottom waters (ca. 2 ‰) (Fig. 5, 10a). The heavier $\delta^{18}O$ of HL and the youngest LL2 is similar to that of recent benthic foraminifera and suggests that they formed in bottom waters having similar physical and chemical characteristics as today with a comparable water-column stratification (Fig. 10a). By contrast, the $\delta^{18}O$ values of older LL2 are similar to those of *G. ruber* formed in Blue Nile-dominated/low-salinity water during the summer, suggesting homogenous temperature and salinity gradients throughout the water-column (Fig. 10a). This observation has several important implications.

The $\delta^{18}O$ of older LL2 around -2 ‰ implies that either LL2 formed in waters that remained dominated by the Nile freshwater input all year long, or that the formation of calcite in LL2 is related to the beginning of the flood season. According to isotopic data from various foraminifera species analysed in a sediment core south of Crete, low salinities typical of large Nile runoff was only a pervasive feature in the Levantine Basin (Tang and Stott, 1993). We therefore propose that the formation of the LL2 likely occurred during the onset of the Nile floods at the end of the spring/beginning of the summer, providing cations to the bicarbonate-rich waters that developed as a result of anaerobic oxidation of organic matter off the Nile mouth (as derived from $\delta^{13}C$ signature of LL2, § 5.2.1) (Fig. 9b). At the end of the dry (winter) period, the Nile River water is indeed enriched in cations ($Ca^{2+}$, $Mg^{2+}$) (Dekov et al., 1997). The supply of cations to bottom waters enriched in $CO_3^{2-}$ ions might have led to the supersaturation of seawater with respect to calcite and its precipitation on sediment particles at the seafloor.

The large range of $\delta^{18}O$ values in LL2 suggests important reorganisation of water masses and water-column stratification during sapropel S1 offshore the Nile River mouth. Low $\delta^{18}O$ during the oldest part of the sapropel indicates that temperature and salinity gradients were homogenous throughout the water column, with water-masses down to 700 m having a Nile freshwater fingerprint (Fig. 10a). As discussed earlier, the formation of authigenic calcite in core P33 likely occurred at the beginning of the summer flood season and the measured $\delta^{18}O$ therefore suggests a deep penetration of the freshwater plume on the Nile DSF. The convective mixing of fresh- and seawater driven by the formation of salt fingers below the flood plume described previously would provide a potential mechanism to draw low-salinity water to bottom parts of the Nile DSF (§ 5.1). It was shown that this process of double-diffusive (diapycnal) mixing enhances vertical fluxes of salt, nutrients and oxygen in the ocean to intermediate depths (Oschlies et al., 2003; Schmitt et al., 2005). We therefore hypothesize that during the deposition of S1, the freshwater released during Nile floods was efficiently transported downwards on the Nile

DSF, reaching water depth up to 700 m. The younger LL2 has a $\delta^{18}O$ value of 0.6 ‰, between the $\delta^{18}O$ of HL and
benthic foraminifera (ca. 1.5 ‰) and the $\delta^{18}O$ of present-day *G. ruber* (ca. 0 ‰). This indicates either a smaller
inflow of freshwater during the Nile floods or a reduced vertical mixing.

**5.3. Spatial and temporal variability of deoxygenation in the SE Levantine Basin during sapropel S1**

Based on the observations of seasonal sublayer deposits and in particular the occurrence of authigenic calcite, we
derived that bottom waters offshore the Nile mouth were anoxic almost all-year round during sapropel S1
deposition and potentially reached euxinic and methane-rich states in spring. This indicates that even though the
Nile flow decreased during the winter-spring months, potentially permitting to form the SIW in the Aegean Sea
(Zirks et al., 2019), the circulation of this intermediate water mass was too sluggish to efficiently ventilate the
basin. During the large summer Nile floods, efficient vertical mixing was probably able to draw low-salinity
nutrient-rich and sediment-laden water masses down to 700 m w-d.
By comparing available markers of deoxygenation in the three sediment cores (in particular the S/Cl records and
occurrence of laminations), we can determine the changes in seawater chemistry at centennial- to millennial-scale
along a bathymetric transect (Fig. 11d). We have added two deeper cores MD04-2726 and MS27PT (retrieved at
1060 and 1390 m w-d, resp.) for which S/Cl ratios were published previously (Ménot et al., 2020; Revel et al.,
2015). The S/Cl records trace the diagenetic formation of sulphur minerals such as pyrite ($FeS_2$), greigite ($Fe_3S_4$)
or pyrrhotite ($Fe_7S_8$) from the pore-water sulphides (Liu et al., 2012; Matthews et al., 2017; Revel et al., 2015).
This process is favoured by oxygen depletion, which leads to the accumulation of dissolved sulphides in the pore-
waters and precipitation of solid iron sulphides. These iron sulphides are generally resistant to post-depositional
oxygenation and therefore permit the identification of past hypoxic intervals that might have be partly reoxidized
(Larrasoaña et al., 2006).

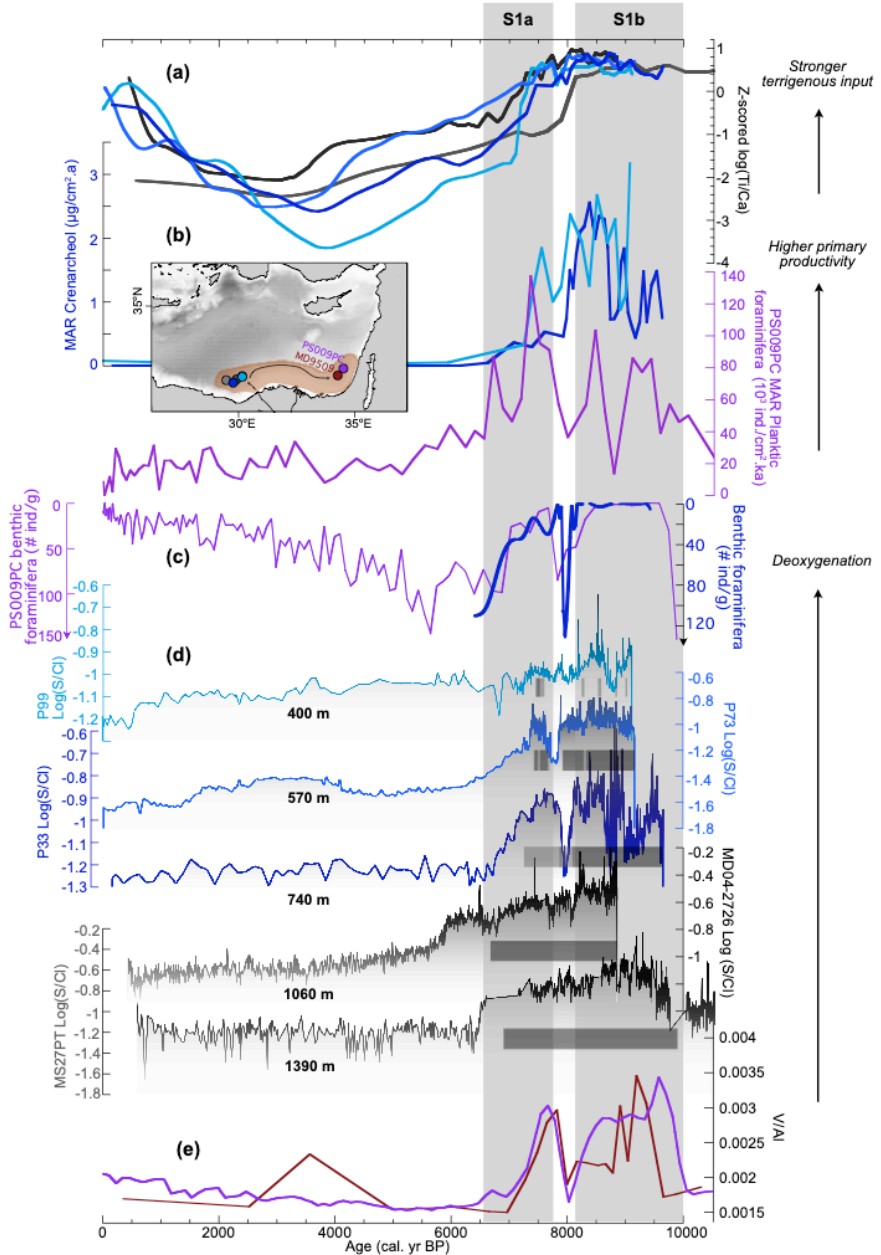


**Figure 11. Regional compilation of changes in terrigenous, primary productivity and oxygenation conditions during the Holocene.**
(a) Standardized (z-score) records of log (Ti/Ca) of cores P99 (light blue), P73 (blue), P33 (dark blue), MD04-2726 (dark grey),
MS27PT (grey) (Revel et al., 2015). Z-scoring (x-mean(Ex)/stdev) allows to compare trends in a similar proxy from different archives
by scaling the y-range and aligning the means on 0. (b) MAR of crenarchaeol in cores P99 (light blue) and P33 (dark blue), planktonic
foraminifera in core PS009PC (purple) (Mojtahid et al., 2015). (c) Abundance of benthic foraminifera in core P33 (cubic spline, dark
blue, inverse y-scale) and core PS009PC (purple). (d) Log(S/Cl) records and lamination patterns in cores from the Nile DSF
(shallower to deeper core site from top to bottom): P99 (light blue), P73 (blue), P33 (dark blue), MD04-2726 (dark grey), MS27PT
(grey) (Revel et al., 2015). (e) V/Al records in cores PS009PC (purple) (Hennekam et al., 2014) and MD9509 (brown) (Matthews et
al., 2017). Grey vertical bars: sapropel S1a and S1b; lighter grey bars: periods of slight reoxygenation; white bar: S1 interruption
(at ca. 8 ka BP). Map in (b) showing the location of the cores along the path of the Levantine Jet.

In order to determine potential forcing factors, this bathymetric and temporal record of paleo-oxygenation will be compared with regional reconstructions of terrigenous input and primary productivity (Fig. 11a,b). The dynamics of terrigenous input and primary productivity are coherent over the western Nile DSF as shown by the good alignment of Ti/Ca ratios measured in all cores (Fig. 11a) and the crenarchaeol fluxes from P33, P99, MS27PT (Ménot et al., 2020; Revel et al., 2015) and GeoB7702-3 (Castañeda et al., 2010) (Fig. 7, 11b). Biomarker records can be affected by post-depositional oxidation, especially by the downward diffusion of oxygen into the reduced sapropelic sediments (i.e., a so-called burn-down process, Rutten and de Lange, 2003). However, comparable trends between biomarker records and other markers of primary productivity from regional archives that are not affected by burn-down processes (i.e., flux of planktonic foraminifera in core PS009PC, Mojtahid et al., 2015) suggest that productivity played a larger role than post-depositional oxidation in modulating the biomarker fluxes in our records (Fig. 11b).

### 5.3.1. Waxing and waning of a hypoxic water-mass during the Holocene

Prior to 10 ka BP, most sediment records from the eastern Mediterranean recorded the existence of oxic conditions as reflected by the presence of epibenthic foraminifera at water depths ranging from 500 to 2000 m (Cornuault et al., 2018; Le Houedec et al., 2020; Schmiedl et al., 2010) as well as low sulphur contents and redox sensitive element ratios (RSER) (Azrieli-Tal et al., 2014; Hennekam et al., 2014; Matthews et al., 2017; Tachikawa et al., 2015). At deep site MS27PT offshore the Nile River mouth, suboxic conditions occurred between 12 and 10 ka BP and were accompanied by higher primary productivity (Ménot et al., 2020). A peak of productivity was also identified in some records from the SE Levantine Basin prior to the onset of S1 deposition at depth similar to MS27PT (>1000 m) (Zirks et al., 2021; Zwiep et al., 2018) but not at shallower depths (Castañeda et al., 2010; Mojtahid et al., 2015).

All cores from the Nile DSF show a pronounced decrease in oxygenation between 10 and 6.5 ka BP, marked by higher S/Cl records and the quasi-absence of benthic foraminifera, followed by relatively stable and oxic conditions in the younger parts of the cores (Fig. 11d). In core P33, we propose that the bottom waters were sulfidic to methanic during S1 deposition, based on $\delta^{13}C$ measurements on LL2 (see § 5.2.1). Such conditions have been detected only in deeper parts of the Levantine Basin during S1 (>2000 m; Azrieli-Tal et al., 2014) and shallower sites located along the path of the Levantine Jet were found to be anoxic but not euxinic between 10 and 7 ka BP (Hennekam et al., 2014; Matthews et al., 2017; Zirks et al., 2021). Recent results hypothesize that high input of iron through enhanced Nile runoff during S1 likely prevented the bottom waters on the coast of Israel (>1000 m w-d) to become sulfidic and instead remained anoxic (Zirks et al., 2021). Our core sites on the Nile DSF are shallower and under

direct influence of fluvial nutrient release compared to the more distal sites of Zirks et al. (2021). Higher primary
productivity and stronger bacterial respiration of sinking organic matter might have been sufficient to balance the
large input of iron- and manganese-rich reactive phases and shift the bottom waters in a euxinic state. In addition,
according to our seasonal reconstructions, euxinia might have developed in the bottom waters during a short
interval in the spring (§5.2.1 and Fig. 9c), which might not be recorded by sampling at centennial-scale resolution.
A complete reoxygenation is achieved at all depths by 6.5 ka BP (and by 6 ka at 1060 m water depth, Revel et al.,
2015). At the scale of the Levantine Basin, the recovery of benthic ecosystems appears to have followed a time-
transgressive pattern, with shallower sites depicting oxic conditions as early as 8 ka BP (Schmiedl et al., 2010).
Such a pattern is also evidenced on the Nile DSF with deoxygenation conditions being maintained until ca. 6.5-6
ka BP at depths >1000 m while shallower sites already started recovering around 7.5 ka BP (Fig. 11d).

## 5.3.2. Dynamic fluctuations in time and space

Centennial fluctuations in oxygenation conditions are observed on the Nile DSF between 570 and 1400 m w-d
during sapropel 1a (10-8 ka BP) (Fig. 11d). Multiple intervals of faint laminations occur in core P99 every 200-
300 yrs (i.e., at 9, 8.8, 8.6, 8.3 ka BP), suggesting a dynamic vertical structure of the upper boundary of the hypoxic
water-mass. In particular, a pronounced shoaling to 400 m w-d is observed around 8.5 ka (Fig. 11d). Cores at
intermediate depth also show variations in the oxygenation state with lower S/Cl records around 9.2 and 8.3 ka BP
recorded in both cores P33 and P73 (although more prominent in P33) and at 8.7 ka BP recorded in core P33 (Fig.
6 and 11c). Millennial-scale variations in bottom-water oxygenation were also recorded at similar depths by nearby
cores PS009PC and 9509 as evidenced by fluctuations in the RSER (Fig. 11e) and in the benthic fauna (Hennekam
et al., 2014; Le Houedec et al., 2020; Matthews et al., 2017). Below 1000 m w-d on the western Nile DSF, anoxic
conditions have likely been more stable during S1a, as suggested by continuously laminated sapropels and
relatively constant S/Cl records in deeper cores MD04-2726 and MS27PT (Fig. 6, 11d). Due to complex diagenetic
processes involving sulphur cycling, the S/Cl may not have recorded rapid changes in the oxygenation state as
shown by RSER (Matthews et al., 2017; Tachikawa et al., 2015). However, the presence of continuous laminations
throughout the sapropel sequence strongly suggests stable anoxic conditions at depths >1000 m. During S1a, wide-
spread anoxia is recorded between 500 and 3000 m w-d in the LB (De Lange et al., 2008; Zirks et al., 2021).
A pronounced but short-lasting interval of reoxidation at ca. 8 ka BP is observed on the western Nile DSF in the
upper 740 m, marked by a lack of laminations at 400 and 570 m depth, faint laminations and recolonization of the
seafloor by benthic foraminifera at 740 m water depth and reduced S/Cl records at all depths (Fig. 7, 11d). The
presence of continuous laminations in the deeper cores indicates that conditions remained anoxic during this time

interval at depths >1000 m. The reoxygenation event at 8 ka BP is a widespread marker in the eastern Mediterranean generally referred to as sapropel S1 interruption, which has been related to a drastic decrease in Nile runoff around the 8.2 ka event (Blanchet et al., 2013; Rohling et al., 2015). In nearby cores located above 900 m water depth at the cost of Israel, the timing of this reoxygenation event was similar (Fig. 11e) (Hennekam et al., 2014; Matthews et al., 2017) but these more distal cores reached a fully oxic state as indicated by the presence of epibenthic fauna (Fig. 11c) (Le Houedec et al., 2020; Schmiedl et al., 2010). Most sites in the Levantine Basin from depths ranging from 500 to 3000 m depict suboxic to oxic conditions during S1 interruption (e.g., Azrieli-Tal et al., 2014; Gallego-Torres et al., 2010; Kuhnt et al., 2008; Mercone et al., 2001; Tachikawa et al., 2015). Low values of bottom-water $\delta^{13}C$ during the S1 interruption, however, indicate stagnating water masses (Schmiedl et al., 2010).

Hypoxic conditions developed again on the western NDSF around 7.8-7.5 ka BP (sapropel S1b) (Fig. 11d). If bottom waters were suboxic at 740 m as indicated by the presence of a few benthic foraminifera and faint laminations in core P33, laminated intervals and high lycopane fluxes in cores P73 and P99 suggest that anoxic conditions existed intermittently between 600 and 400 m water depth during S1b (Fig. 6a, 11d). At deeper sites, conditions remained anoxic until ca. 6.7 ka BP when laminations stopped in core MS27PT (Fig. 11d). This suggests the existence of split anoxia, during which both shallower and deeper water masses were deprived of oxygen due to the respiration of rapidly-sinking organic detritus, while intermediate water masses remained suboxic (Bianchi et al., 2006; Rush et al., 2019). In most cores from the Levantine Basin, suboxic to anoxic conditions were re-established by 7.8 ka BP but generally lasted only a few hundred years and were rather unstable (Fig. 11c, e) (Schmiedl et al., 2010). Suboxic conditions and a gradual recolonization of the seafloor by benthic foraminifera indicating a gradual reoxygenation between 7.5 and 6.5 ka BP were observed at several sites along the Levantine Jet pathway at depths <1000 m (Fig. 11c) (Le Houedec et al., 2020; Schmiedl et al., 2010).

### 5.3.3. The role of runoff-driven eutrophication in the SE Levantine Basin

Recent modelling experiments indicated that the formation of a basin-wide oxygen depletion in the deep Mediterranean Sea during sapropel S1 required a multi-millennial deep-water stagnation (Grimm et al., 2015). It was recently determined that the increasing sea-level during the deglaciation allowed the inflow of fresher Atlantic-derived water, thereby increasing the buoyancy of surface waters and decreasing deep-water ventilation in the eastern Mediterranean (Cornuault et al., 2018). This initial stagnation may have been reinforced by larger inputs of freshwater from the Nile river, which enhanced water-column stratification from 14 ka BP onwards (Castañeda et al., 2016; Cornuault et al., 2018; Vadsaria et al., 2019).

The marked onset of anoxia in the eastern Mediterranean Sea at ca. 10 ka BP is striking as it is recorded uniformly
at various depths in the basin (De Lange et al., 2008; Schmiedl et al., 2010). Detailed faunal and isotopic studies
showed that deep-water stagnation gradually developed from 17 ka BP onwards, with a distinct drop in $\delta^{13}C$ around
11 ka BP (Cornuault et al., 2018; Schmiedl et al., 2010). It was suggested that conditions remained oligotrophic
until 10 ka BP, allowing benthic fauna to dwell even in stagnating water masses, after which benthic ecosystems
passed a threshold that led to the development of anoxic conditions (Abu-Zied et al., 2008; Kuhnt et al., 2008). In
deeper parts of the LB, the development of anoxia might primarily result from long-term preconditioning of water
masses as modelled by Grimm et al. (2015). However, this modelling experiment does not produce anoxic
conditions in water depths above 1800 m, which suggests that other forcing factors might drive deoxygenation
dynamics at intermediate depths. As seen in section 5.3.2, considerable spatial and temporal variability in
oxygenation was observed in the SE Levantine Basin. We propose here that these changes were largely driven by
modulation of runoff-derived nutrient loadings. Around 10 ka BP onwards, a drastic increase in primary
productivity has been identified in the SE Levantine Basin (Fig. 11b) (Mojtahid et al., 2015), which coincides with
high precipitation and runoff in the Nile River basin (Castañeda et al., 2016; Weldeab et al., 2014). It is therefore
postulated that the large nutrient loading delivered by summer floods of the Nile River led to a switch from an
oligotrophic to a meso- or eutrophic state in the SE Levantine Basin, with high levels of primary productivity in
surface waters. As an extended version of the Nile blooms forming offshore the Nile mouth during historical times
(Halim et al., 1967), peak Nile runoff during the mega-summer monsoon of the African Humid Period probably
fertilised large parts of the SE Levantine Basin during sapropel S1 deposition (Hennekam et al., 2015; Schmiedl et
al., 2010). Lower oxygenation indices in cores located closer to the Nile River as shown in our study further support
the pivotal impact of Nile River-induced eutrophication on the rapid spread of hypoxia in the SE Levantine Basin
(Schmiedl et al., 2010; Zirks et al., 2019). Moreover, it was recently proposed that rapid changes in oxygenation
could occur within 200-500 years for water masses located between 500 and 1800 m water depth due to oxygen
utilization by bacterial organic matter degradation (Zirks et al., 2019). We therefore propose that most of the
changes in the strength and extent of hypoxia offshore the Nile River were related to runoff-induced fertilisation
and subsequent eutrophication of the water-column. However, these centennial- to millennial-scale variations were
superimposed on the multi-millennial development of deep-water stagnation and suggest complex interactions and
feedback processes between deep-water circulation, stagnation/stratification and eutrophication for driving
deoxygenation dynamics. The rapid deoxygenation of the SE Levantine Basin around 10 ka BP following
widespread eutrophication of surface waters, after thousands of years of deep-water stagnation, also calls for
modelling experiments to include temporal variations in nutrient loading and river runoff, which are important
variables in the present Nile coastal system (Nixon, 2003; Powley et al., 2016).

## 6 Conclusions

By combining microfacies analyses with downcore geochemical measurements partly at seasonal resolution, our
study provides a first estimation of changes in oxygenation conditions of the bottom waters on the western Nile
deep-sea fan in the SE Levantine Basin. The regular seasonal alternation of detrital, biogenic and chemical
sublayers in the laminated sequence deposited during sapropel S1 in a set of cores from different water depths on
the Nile deep-sea fan are ascribed to seasonal changes (Fig. 9b). Strong summer Nile floods during S1 led to the
deposition of thick (up to a few mm) silt-sized detrital sublayers that dominate total layer thickness. The large
floods likely triggered surface blooms of phyto- and zooplankton in autumn similar to historical "Nile blooms"
(Halim et al., 1967). The subsequent deposition of clay-rich detrital sublayers was associated with the low discharge
regime of the Nile during winter. The reduction of Nile runoff might have sufficiently decreased the surface water
salinity in the Levantine Basin, thereby allowing the formation of a sluggish intermediate water mass in the Aegean
Sea (Zirks et al., 2019). The occurrence of inorganic carbonate sublayers in several laminated cores from the
western Nile deep-sea fan suggests that bottom waters reached a supersaturation state for calcite. The depleted $\delta^{13}$C
signature of these sublayers points to the existence of anoxic to euxinic (and sometimes methane-rich) bottom
waters accompanied by a high level of anaerobic remineralisation of organic matter leading to high alkalinity. The
most likely process initiating the deposition of these layers was the onset of the Nile floods triggered by increased
precipitation, which supplied sufficient amounts of cations to the seawater. First measurements of individual sub-
millimetre layers underpin their potential to reconstruct seawater chemistry at times when no benthic fauna existed.
On millennial time-scales, we show that variations in oxygenation dynamics at intermediate depth in the SE
Levantine Basin followed changes in primary productivity driven by nutrient fertilisation during high Nile runoff.
Deoxygenation above the Nile deep-sea fan shoaled to water depths as shallow as 400 m and varied on centennial-
to millennial-scales in the upper 750 m of the water column. In contrast, the records of cores located below 1000
m water-depth reflect more stable anoxic conditions between 10 and 6.5 ka BP. The development and fluctuations
of anoxic conditions during sapropel S1 are coherent along the path of the Levantine Jet suggesting a common
control. The posited multi-millennial development of deep-water stagnation was likely a prerequisite to trigger
anoxia in the deeper parts of the eastern Mediterranean Basin (Cornuault et al., 2018; Grimm et al., 2015), but
changes in primary productivity in surface waters probably drove the rapid changes in oxygenation state of
intermediate water masses in the SE Levantine Basin through eutrophication processes. Indeed, tight temporal links

between regional productivity records, oxygenation and runoff dynamics, as well as the evidence of stronger deoxygenation close to the Nile mouth point to a pivotal role of runoff-driven nutrient supply and fertilisation. Furthermore, the rapid switch towards anoxic conditions around 10 ka BP suggests the crossing of thresholds or tipping points, which still remain elusive. Such processes could be further explored using transient modelling experiments incorporating variations in nutrient loading and Nile runoff at various time-scales for understanding feedback processes and system sensitivity.

**Data availability**

All data presented in this paper will be available at the PANGEA database upon publication of the paper and for the reviewers

Table S1: Elemental contents measured by XRF for core P362/2-33 and shown in Fig. 2, 6, 8 and 9.

Table S2: Elemental contents measured by XRF for core P362/2-99 and shown in Fig. 2, 6, 8 and 9.

Table S3: Elemental contents measured by XRF for core P362/2-73 and shown in Fig. 2, 6, 8 and 9.

Table S4: Oxygen and carbon isotope data for core P362/2-33 shown in Fig. 4.

Table S5: Biomarker data for core P362/2-33 shown in Fig. 6, 7, 8 and 9.

Table S6: Biomarker data for core P362/2-99 shown in Fig. 6, 7, 8 and 9.

**Supplement Link**

For the present submission, supplementary figures are provided in a separate file.

**Author contribution**

CLB designed the study, measured and analysed the data. RT measured and analysed the XRF elemental contents. AES assisted with XRD measurements. SS, MF and AB provided guidance, lab space and logistic support for biomarker measurements, sedimentology and micro-facies analyses, respectively. CLB wrote the manuscript, to which all co-authors contributed.

**Competing interests**

The authors declare that they have no conflicts of interest.

## Acknowledgements

Measurements for this paper were carried out during a DFG own-position grant to C.B. at GEOMAR (Kiel, Germany) and NIOZ (Texel, Netherlands) (BL2111/1-1, 2010-2013) and during a reintegration grant at GFZ (Potsdam, Germany) (2018-2020). We are grateful to our colleagues from section 4.3 at GFZ for their support in term of sample preparation (B. Brademann), measurements (Stable isotopes lab, B. Plessen and S. Pinkerneil) and discussions (esp. J. Mingram). We also acknowledge G. Ménot, M. Mojtahid, R. Hennekam, G. Bayon and A.Matthews who kindly provided and discussed their data. The paper was written during the lockdown due to COVID-19 (March to June 2020) with background support from Daniel Hope and Max Richter.

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
