# Peer review of "Deoxygenation dynamics on the western Nile deep-sea fan during"

_Climate of the Past, 2020_

## Referee Comment (RC1) · Anonymous Referee #1 · 26 Oct 2020

The authors present high-resolution sedimentological (microfacies) and geochemical data records for three upper bathyal sediment cores from high-accumulation sites offshore the western distributaries of the Nile River delta. The sediment cores encompass the Holocene period including laminated sediments, which have been deposited during the last stagnation period of sapropel S1 in the Eastern Mediterranean Sea (EMS). The temporal resolution of the study region is unprecedented and allows for resolving even seasonal changes and thus the characterization of changes in the strength of the yearly Nile flood and associated regional environmental conditions at the core sites. In this reconstruction, the application of microfacies analyses, previously established for laminated lake sediments provides essentially novel information on the paleohydrology

of the Nile River and related summer floods. The manuscript is generally well designed and follows a widely comprehensive argumentation. Nevertheless, the following issues should be considered in the revision process:

1) I disagree with the general conclusion concerning the role of productivity as a major driver in S1 formation in the entire EMS basin. Enhanced productivity likely plays an important role during S1 deposition in the vicinity of the Nile delta and in the eastward flowing surface water current along the Israeli coast. Eutrophication of surface waters has been documented in sediments retrieved from the Israeli continental margin. However, evidence from abyssal and bathyal core sites from the central parts of the basins and from the northern margins of the EMS provide a controversial picture, in many cases lacking evidence for eutrophication during S1 formation.

In this context, the manuscript would profit from a more critical discussion of preservation issues with respect to the interpretation of geochemical proxies since syn- or post-depositional degradation of organic compounds may play a significant role. For example, the study of Moebius et al. (2010, Biogeosciences 7, 3901-3914) suggested that "... preservation plays a major role for the accumulation of organic-rich sediments casting doubt on the need of enhanced primary production for sapropel formation." In addition, the study of Grimm et al. (2015) used a sophisticated regional ocean circulation model coupled to a biogeochemical model to explore the potential role of productivity and associated organic carbon fluxes in S1 formation (this is exactly what you ask for at the end of your conclusions!). They found that the required oxygen depletion is mainly depending on the stratification intensity and stagnation history of the water column and that a basin-wide productivity increase is not required as a prerequisite for sapropel formation.

These and other results suggest that the observations of upper bathyal anoxia off the Nile delta may not be representative for environmental changes in the entire basin but may rather represent a regional situation due to its vicinity to the Nile nutrient source. This suggestion is also confirmed by the restriction of sustained basin-wide

early Holocene anoxia to water depths below approximately 1500 m (De Lange et al., 2008, Nature Geoscience, 1: 606–610) while sustained anoxia seem to have prevailed at shallower water depths under the direct influence of Nile outflow waters. In the revision of the manuscript, the authors should acknowledge the still controversial findings and diverse evidence from both proxy and model applications. In doing so, I suggest to abandon conclusions drawn for the entire EMS (including Figure 10) and rather focus on the Nile river dynamics and regional environmental impacts. These results are novel enough and provide a variety of essentially new insights into past hydrological changes and marine depositional responses.

2) You argue that the Nile flood triggered blooms of planktonic foraminifera and calcareous nannoplankton in autumn, similar to natural Nile blooms prior to the construction of the Aswan dam. Typically, eutrophication in surface waters influenced by riverine nutrient input results in diatom blooms as described for the historical Nile floods (Halim et al., 1967). To date, Halim et al. (1967) do not mention planktonic foraminifera and coccolithophorids as typical groups responding to the seasonal nutrient input. In the modern EMS, planktonic foraminifera and coccolithophorids are widely distributed and even thrive in the ultraoligotrophic parts of the basin as in many other oligotrophic oceans. Did you observe any layers rich in opal? In the EMS, opal remains are often not preserved, which may inhibit documentation of the proper succession of nutrient-driven phyto- and zooplankton association in the sediment. Your discussion and interpretation should be more specific here, acknowledging a correct assessment of plankton response to river-induced surface water eutrophication.

3) The presence of benthic foraminifera (BF) in the interruption and upper part of S1 clearly indicates the absence of permanent anoxic conditions and at least intermittently oxygenated time periods at the deepest core site (738 m water depth) since around 8200 years B.P. It is a pity that no BF data are available for the other two cores. On the other hand, the presented data on the genus level do not provide sufficient insights into the BF fauna and thus benthic ecosystem state since different species of the same

genus often have contrasting ecology. In addition, proper taxonomy is required for excluding the potential effects of down-slope transport. The authors mention the genus Cibbides(?). I guess you refer to the genus Cibicides or Cibicidoides, do you? Species of the genus Cibicides commonly inhabit shelf environments in the EMS and thus would represent reworked tests when found on the continental slope. On the other hand, Cibicidoides is an autochthonous deep-sea taxon. In order to avoid misinterpretations, it would be more honest to lump all species together and present data on the BF presence/absence or concentration (e.g. individuals per g dry sediment).

4) I wonder if you have observed any gypsum crystals in the sediment. Post-depositional oxidation of iron sulfides and calcium carbonate often result in precipitation of gypsum in sapropelic sediments.

5) Some of the figures appear too busy and the blueish colors used do not contrast sufficiently. This is particularly visible in figures 6, 8 and 9. I suggest using more contrasting colors and also avoid overlying symbols and lines.

I hope that my suggestions prove useful for the revision of the manuscript!

---

## Short Comment (SC1) · 4 Nov 2020

Tjallingii et al. (2007) is cited in line 193, but does not appear in the reference list.

---

## Author Comment (AC1) · 4 Nov 2020

Dear Mr Burke,

Thanks a lot for sending us the notice for the missing reference. We will correct that in the final manuscript but here is the citation:

Tjallingii, R., Röhl, U., Kölling, M. and Bickert, T.: Influence of the water content on X-ray fluorescence core-scanning measurements in soft marine sediments, Geochemistry, Geophysics, Geosystems, 8(2), doi:10.1029/2006GC001393, 2007.

Best regards, Cecile Blanchet, on behalf of all co-authors.

---

## Referee Comment (RC2) · Anonymous Referee #2 · 19 Nov 2020

Applying multiproxy approach (micro-facies analysis with inorganic and organic geo-chemistry) to three cores of high sedimentation rates from the Nile deep-sea fan, Blanchet et al. reconstructed seasonal to millennial-scale variability of detrital inputs, biological productivity and bottom/pore water oxygenation states during the Holocene covering sapropel S1 deposition period. The highly resolved records provide very useful information on the impact of Nile flooding as both freshwater and nutrient supplier. The quality of data is high and scientific subject fits well the field covered in Climate of the Past. I, however, have several concerns that should be solved during the revision process. I develop the points below.

1. Local vs. regional impact of Nile discharge: circulation and productivity The most innovative results of the study are (i) annual cycle of sediment deposition during S1 period in the western Nile deep-sea fan and (ii) additional evidence for heterogeneous bottom/pore water oxygenation states in the coastal regions during S1. Point (i) is well described whereas the transition from (i) to (ii) is abrupt and some key points of (ii) are missing. The most exciting but not fully addressed subject is how the identified annual cycle of Nile discharge and the productivity changes was interacted with basin-scale water circulation and oxygenation states. The authors briefly proposed the impact of nutrient supply by Nile river discharge in relation to the mega-summer monsoon of the AHP by analog with "Nile bloom". However, there exists a growing body of evidences for the leading role of stagnant circulation that pre-conditioned the S1 deposition based on proxy reconstruction and a numerical modeling (Grimm et al., 2015). Since the authors provided unpreceded resolution data, it could be possible to revisit the role of the Nile at finer timescale. For instance, the authors proposed that fine-grained clay-rich particle deposition in laminated layers as a sign of stagnant circulation. How the stagnant circulation was realized and what was the relationship between the observed slower circulation in the Nile deep-sea fan with the ventilation of the other parts of the Levantine Sea? In general, the present manuscript did not sufficiently describe the role of the circulation to the sapropel formation. The present-day circulation pattern is very shortly shown in Section 2 (Regional context) and there is no statement about water mass occupies the three core sites and how the mass is oxygenated under the present condition. This information should be added. Besides, the sketches of an annual cycle shown in Fig. 5c are difficult to understand. What is the size of geographical extension of area for the proposed sedimentation processes?

To clarify the local vs. regional impact of Nile discharge, I would suggest reorganize section 5.3. Do the titles indicated on lines 574, 601 and 653 correspond to the subsection of 5.3, thus the section extends for 7 pages? It is better to start by coastal regions (Nile deep sea fan and Israeli coast) and extend the discussion to the Levantine basin. The authors should be careful about the terms of geographic definition. For instance,

the whole Eastern Mediterranean Sea, including the Adriatic Sea and Ionian Sea, is not treated in the present manuscript, therefore it is inappropriate to use this term in the section title.

2.Do the reconstructions indicate bottom or pore water oxygenation conditions? It is necessary to clarify whether the geochemical signals represent bottom or pore water conditions. Based on the low d13C values of authigenic carbonates from the HL layers during S1 deposition period, the authors proposed anoxic bottom water condition at site P33 (740 m water depth) and possibly at site P73 (570 m water depth). These sites are much shallower than the previously reported anoxic sites during S1 (1000 m and 2000m) based on authigenic carbonates (Aloisi et al., 2002; Bayon et al., 2013). It is not clear what the authors observed is pore water or water column oxygenation state. It is possible that the reconstructed oxygenation state was very localized with patchy distribution that is not suitable to generalize the whole basin. About d18O values of LL2 carbonates, the authors proposed that "temperature and salinity gradients were homogenous throughout the water column". Does it mean that no density gradient existed during authigenic carbonate precipitation in 740 m water column? Is the hypothesis of the homogeneous water column consistent with oxygen depleted bottom water? If so, why the shallowest core (P99) from 400 m water depth showed less lamination despite and higher or comparable alkenone flux than at the deeper core (P33, Fig. 7)?

At last, I have a comment on the general structure of the manuscript. The authors mixed result presentation and some interpretation in section 4 then more detailed discussion followed in section 5. Personally, I prefer to separate result and discussion to avoid mixed objective and subjective descriptions and redundant statements. I understand that this organization is due to multi-proxy reconstructions that necessitate to explain the meaning of numerous proxies. To overcome this complication, the authors may present the tool box classified into target variables before the result section.

I recommend to accept this work after major revision.

Minor / specific comments Line 28 "the entire Levantine Basin". See my comment 1.

Line 48 "decreased" should be replaced by "increased".

Lines 51-52. " the consequently low or quasi-absent primary productivity (Krom et al., 2014)". Delete "or quasi-absent". Pujo-Pay et al. (2011) is probably more appropriate to cite.

Line 59. "the 1980's" should be "the 1980s".

Line 67. Somot et al., 2018. This paper treats the circulation in the NW Mediterranean Sea, not in the Eastern Mediterranean Sea. Adolff et al. (2015) is more suitable to this sentence because the circulation of the whole Mediterranean Sea is studied.

Line 72. "Sapropels have proven a valuable laboratory". This sentence is strange and should be revised.

Line 186. "Aavatech" should be "Avaatech".

Line 218. "trialkyl" or "dialkyl"?

Lines 204-212 and Fig. 4. Indicate the core name of which stable isotope data were obtained.

Lines 282-284. The occurrence of HL in both cores P33 and P73. Is the appearance of HL synchronous for the two cores to support the hypothesis of reduced oxygenation state at intermediate water depths?

Line 367. Replace "contents" by "fluxes".

Line 375. Add the corresponding water depths of the indicated cores after "deeper cores MS27PT and GeoB7702-3".

Line 385. Add "variability" after "runoff".

Line 400 and Fig. 8a,e,h. Why Ti-enrichment occurred at the Late Holocene?

Line 405. Add "ka BP" after "7.2".

Line 406. "Ti/K" should be "K/Ti".

Line 409 and Fig. 8. Indicate the sapropel interval in Fig. 8.

Lines 436-438 and Fig. 3c. It is not clear that smectite, plagioclase and iron-titanium oxides are more abundant in DL2 relative to the other layers. According to the figure caption, Fe-rich phases are probably pyrite, not iron-titanium oxides.

Line 459. Add a reference after "interface".

Lines 473-475. The authors may cite a recent regional modelling study that simulated surface salinity anomaly distribution in the Mediterranean Sea by increasing Nile river discharge (Vadsaria et al., 2019).

Lines 476-479. It is unclear whether the authors treated here the mixing of water masses or sedimentation processes. Please clarify.

Lines 495-496. Add the reference for the bottom water d13C value of -7‰ off the Nile river mouth.

Lines 511-514. About d18O of authigenic carbonates. What does "theoretical d18O" mean? D18O value of equilibrated calcite? If so, how do you estimate temperature and seawater d18O of surface and bottom waters? Lines 550-564. Delete the description corresponding to K. The results are already shown and K/Ti is not presented in Fig. 9a.

Lines 638-641. SIW was already defined.

Numbering of figures. The authors used (a), (b), (c)... to describe the curves of reconstruction instead labelling different panels (ex. Figs. 6, 7 and 8). Sometimes the different curves are combined and a common label is used. The lamination patterns are also labelled. This presentation is confusing. The authors may label the different panels or number all curves. The lamination patterns can be shown without numbering.

Fig. 3. Please indicate which results correspond to which core (core P33 or core P73).

Fig. 5a, b. The x, y- axis and axis titles are too small.

Fig. 8 caption. There are errors of numbering. Please check it.

Fig. 9. A section map showing the site positions with the bathymetry will be helpful.

Fig. 10. Use different symbols to distinguish proxies used to evaluate the oxygenation states. Does each panel show the mean state of each period? What is the age rage of the interruption? Did the authors recalibrate the age of the different cores using the same calibration?

References Adloff, F., Somot, S., Sevault, F., Jordà, G., Aznar, R., Déqué, M., Herrmann, M., Marcos, M., Dubois, C., Padorno, E., Alvarez-Fanjul, E., and Gomis, D.: Mediterranean Sea response to climate change in an ensemble of twenty first century scenarios, Clim. Dyn., doi: 10.1007/s00382-015-2507-3, 2015. 1-28, 2015.

Pujo-Pay, M., Conan, P., Oriol, L., Cornet-Barthaux, V., Falco, C., Ghiglione, J. F., Goyet, C., Moutin, T., and Prieur, L.: Integrated survey of elemental stoichiometry (C, N, P) from the western to eastern Mediterranean Sea, Biogeosciences, 8, 883-899, 2011.

Vadsaria, T., Ramstein, G., Dutay, J. C., Li, L., Ayache, M., and Richon, C.: Simulating the Occurrence of the Last Sapropel Event (S1): Mediterranean Basin Ocean Dynamics Simulations Using Nd Isotopic Composition Modeling, Paleoceanography and Paleoclimatology, 34, 237-251, 2019.

---

## Author Comment (AC2) · 23 Dec 2020

The authors present high-resolution sedimentological (microfacies) and geochemical data records for three upper bathyal sediment cores from high-accumulation sites off-shore the western distributaries of the Nile River delta. The sediment cores encompass the Holocene period including laminated sediments, which have been deposited during the last stagnation period of sapropel S1 in the Eastern Mediterranean Sea (EMS). The temporal resolution of the study region is unprecedented and allows for resolving even seasonal changes and thus the characterization of changes in the strength of the yearly Nile flood and associated regional environmental conditions at the core sites. In this reconstruction, the application of microfacies analyses, previously established for laminated lake sediments provides essentially novel information on the paleohydrology of the Nile River and related summer floods. The manuscript is generally well designed and follows a widely comprehensive argumentation.

Thank you for these supportive comments.

Nevertheless, the following issues should be considered in the revision process:

1) I disagree with the general conclusion concerning the role of productivity as a major driver in S1 formation in the entire EMS basin. Enhanced productivity likely plays an important role during S1 deposition in the vicinity of the Nile delta and in the eastward flowing surface water current along the Israeli coast. Eutrophication of surface waters has been documented in sediments retrieved from the Israeli continental margin. However, evidence from abyssal and bathyal core sites from the central parts of the basins and from the northern margins of the EMS provide a controversial picture, in many cases lacking evidence for eutrophication during S1 formation. In this context, the manuscript would profit from a more critical discussion of preservation issues with respect to the interpretation of geochemical proxies since syn- or post-depositional degradation of organic compounds may play a significant role. For example, the study of Moebius et al. (2010, Biogeosciences 7, 3901-3914) suggested that "... preservation plays a major role for the accumulation of organic-rich sediments casting doubt on the need of enhanced primary production for sapropel formation." In addition, the study of Grimm et al. (2015) used a sophisticated regional ocean circulation model coupled to a biogeochemical model to explore the potential role of productivity and associated organic carbon fluxes in S1 formation (this is exactly what you ask for at the end of your conclusions!). They found that the required oxygen depletion is mainly depending on the stratification intensity and stagnation history of the water column and that a basin-wide productivity increase is not required as a prerequisite for sapropel formation. These and other results suggest that the observations of upper bathyal anoxia off the Nile delta may not be representative for environmental changes in the entire basin but may rather represent a regional situation due to its vicinity to the Nile nutrient source. This suggestion is also confirmed by the restriction of sustained basin-wide early Holocene anoxia to water depths below approximately 1500 m (De Lange et al.,2008, Nature Geoscience, 1: 606–610) while sustained anoxia seem to have prevailed at shallower water depths under the direct influence of Nile outflow waters. In the revision of the manuscript, the authors should acknowledge the still controversial findings and diverse evidence from both proxy and model applications. In doing so, I suggest to abandon conclusions drawn for the entire EMS (including Figure 10) and rather focus on the Nile river dynamics and regional environmental impacts. These results are novel enough and provide a variety of essentially new insights into past hydrological changes and marine depositional responses.

We thank Reviewer #1 for this crucial comment. The interplay between productivity, preservation and stagnation is a complex issue and we acknowledge that it is probably wiser to draw conclusions for the environments "closer to home", i.e., directly under the influence of the Nile sediment plume. In the revision of the manuscript, we will include a clearer discussion of preservation issues and the role of seawater stagnation. We will also discuss our findings as being relevant for the Nile deep-sea fan and Israeli coast and remove Figure 10, as requested.

However, please note that our point here was not to refute the role of long-term stagnation on the development of basin-scale anoxia (indeed very clearly demonstrated by model results of Grimm et al. (2015) and backed by proxy-data of Cornuault et al. (2018) -both repeatedly cited in the manuscript), but rather to discuss the additional role played by variations in primary productivity on shorter time scales. That enhanced productivity (and resulting eutrophication) might not be sufficient to explain deoxygenation is clear, but the role of changes in productivity and freshwater release on centennial-scale changes in oxygenation has not been explored by Grimm et al. (2015) ( who attributed shorter time scale ventilation events to cold events producing denser waters). Last but not least, the model of Grimm et al. (2015) does not lead to the development of anoxia in water masses above 1800 m in the Levantine Sea (nor in the Ionian Sea), the occurrence of which has, however, been demonstrated by our data and other studies. Hosing experiment by Vadsaria et al (2019) recently challenged the results of Grimm et al. (2015) concerning the role of freshwater release by the Nile at higher resolution but their model also does not explore the consequences of transient changes in water-mass structure, freshwater and nutrient release for centennial-scale changes in oxygenation. At present, there is no model able to look at this succession of events in a more dynamic manner.

Organic matter preservation is indeed responsible for parts of the variability in our record (e.g., lycopane is a purely preservation-driven record), but cannot account for the changes in planktonic foraminifera accumulation rates observed on the Israeli Coast, redrawn in Fig. 9b (Mojtahid et al., 2015).

We will take into account this important comment and focus our revised manuscript on building a picture of changes occurring along the path of the Nile sediment plume. If we are able to reconstruct past oxygenation dynamics for the Nile plume, the question remains indeed to what extent these processes have influenced other locations in the Levantine Basin.

2) You argue that the Nile flood triggered blooms of planktonic foraminifera and calcareous nannoplankton in autumn, similar to natural Nile blooms prior to the construction of the Aswan dam. Typically, eutrophication in surface waters influenced by riverine nutrient input results in diatom blooms as described for the historical Nile floods (Halim et al., 1967). To date, Halim et al. (1967) do not mention planktonic foraminifera and coccolithophorids as typical groups responding to the seasonal nutrient input. In the modern EMS, planktonic foraminifera and coccolithophorids are widely distributed and even thrive in the ultraoligotrophic parts of the basin as in many other oligotrophic oceans. Did you observe any layers rich in opal? In the EMS, opal remains are often not pre-served, which may inhibit documentation of the proper succession of nutrient-driven phyto- and zooplankton association in the sediment. Your discussion and interpretation should be more specific here, acknowledging a correct assessment of plankton response to river-induced surface water eutrophication.

Reviewer #1 is correct: Halim et al. (1967) did not specifically report carbonate zooplankton for historical floods of the Nile. We will revise this part to be more precise, acknowledge assumptions and provide additional information. Unfortunately, we did not observe any diatoms or opal in the sediments, which, as Reviewer #1 said, probably results from poor opal preservation. However, the deposition of foraminifera and coccoliths layers is associated with distinct layers of organic matter (see image below as an example) and we therefore assume that they represent post-flood layers

following a fertilisation process similar to that occurring during the Nile blooms described by Halim et al. (1967).

[Figure]

3) The presence of benthic foraminifera (BF) in the interruption and upper part of S1clearly indicates the absence of permanent anoxic conditions and at least intermittently oxygenated time periods at the deepest core site (738 m water depth) since around8200 years B.P. It is a pity that no BF data are available for the other two cores. On the other hand, the presented data on the genus level do not provide sufficient insights into the BF fauna and thus benthic ecosystem state since different species of the same genus often have contrasting ecology. In addition, proper taxonomy is required for excluding the potential effects of down-slope transport. The authors mention the genus Cibbides(?). I guess you refer to the genus Cibicides or Cibicidoides, do you? Species of the genus Cibicides commonly inhabit shelf environments in the EMS and thus would represent reworked tests when found on the continental slope. On the other hand, Cibicidoides is an autochthonous deep-sea taxon. In order to avoid misinterpretations, it would be more honest to lump all species together and present data on the BF presence/absence or concentration (e.g. individuals per g dry sediment).

Thanks for this comment and spotting this mistake. Indeed, there was a typo in the text: we have identified specimens of Cibicidoides. We agree that a full investigation of benthic foraminifera would be extremely valuable for these cores to get a more precise picture of bottom-water environments during Sapropel S1 at the Nile mouth. As suggested by Reviewer #1, we will combine the specimens of different genera and show the BF data as individuals/g sediment.

4) I wonder if you have observed any gypsum crystals in the sediment. Post-depositional oxidation of iron sulfides and calcium carbonate often result in precipitation of gypsum in sapropelic sediments.

No gypsum was observed in this core, which might be due to the relatively low Corg content (±1%) compared to other sapropel layers (our cores rather represent sapropelites *sensu stricto*).

5) Some of the figures appear too busy and the blueish colors used do not contrast sufficiently. This is particularly visible in figures 6, 8 and 9. I suggest using more contrasting colors and also avoid overlying symbols and lines.

The figures, esp. the colour codes, will be modified in order to improve readability.

I hope that my suggestions prove useful for the revision of the manuscript!

Yes, they are, thank you very much!

---

## Author Comment (AC3) · 23 Dec 2020

Reply to anonymous Referee #2

Reviewer #2's original comments are in black and our replies are given in blue.

Applying multiproxy approach (micro-facies analysis with inorganic and organic geo-chemistry) to three cores of high sedimentation rates from the Nile deep-sea fan, Blanchet et al. reconstructed seasonal to millennial-scale variability of detrital inputs, biological productivity and bottom/pore water oxygenation states during the Holocene covering sapropel S1 deposition period. The highly resolved records provide very useful information on the impact of Nile flooding as both freshwater and nutrient supplier. The quality of data is high and scientific subject fits well the field covered in Climate of the Past. I, however, have several concerns that should be solved during the revision process. I develop the points below.

Thank you for this positive assessment.

1. Local vs. regional impact of Nile discharge: circulation and productivity The most innovative results of the study are (i) annual cycle of sediment deposition during S1period in the western Nile deep-sea fan and (ii) additional evidence for heterogeneous bottom/pore water oxygenation states in the coastal regions during S1. Point (i) is well described whereas the transition from (i) to (ii) is abrupt and some key points of (ii) are missing. The most exciting but not fully addressed subject is how the identified annual cycle of Nile discharge and the productivity changes was interacted with basin-scale water circulation and oxygenation states. The authors briefly proposed the impact of nutrient supply by Nile river discharge in relation to the mega-summer monsoon of the AHP by analog with "Nile bloom". However, there exists a growing body of evidences for the leading role of stagnant circulation that pre-conditioned the S1 deposition based on proxy reconstruction and a numerical modeling (Grimm et al., 2015). Since the authors provided unpreceded resolution data, it could be possible to revisit the role of the Nile at finer timescale. For instance, the authors proposed that fine-grained clay-rich particle deposition in laminated layers as a sign of stagnant circulation. How the stagnant circulation was realized and what was the relationship between the observed slower circulation in the Nile deep-sea fan with the ventilation of the other parts of the Levantine Sea? In general, the present manuscript did not sufficiently describe the role of the circulation to the sapropel formation. The present-day circulation pattern is very shortly shown in Section 2 (Regional context) and there is no statement about water mass occupies the three core sites and how the mass is oxygenated under the present condition. This information should be added. Besides, the sketches of an annual cycle shown in Fig. 5c are difficult to understand. What is the size of geographical extension of area for the proposed sedimentation processes?

We thank Reviewer #2 for this important comment. In the revised manuscript, we will provide more information on the present-day water-mass structure to the best of our knowledge. In addition, we will review the state-of-the-art on sea water circulation during sapropel deposition. Reviewer #1 suggested to focus on deriving paleoenvironmental information from archives located along the path of the Nile sediment plume, i.e., the Nile deep-sea fan and Israeli coast. We will therefore use our new records to draw a finer picture of seasonal variations on the Nile deep-sea fan and improve the sketches of Fig. 5c (with a better scaled representation). The interesting point made by Reviewer #2 about the possibility to derive further-reaching interpretations about water-mass structure and circulation from our seasonal record will be explored. Generally speaking, we will aim to better exploit the new results provided by the seasonal record.

To clarify the local vs. regional impact of Nile discharge, I would suggest reorganize section 5.3. Do the titles indicated on lines 574, 601 and 653 correspond to the subsection of 5.3, thus the section

extends for 7 pages? It is better to start by coastal regions (Nile deep sea fan and Israeli coast) and extend the discussion to the Levantine basin. The authors should be careful about the terms of geographic definition. For instance, the whole Eastern Mediterranean Sea, including the Adriatic Sea and Ionian Sea, is not treated in the present manuscript, therefore it is inappropriate to use this term in the section title.

Section 5.3, as well as other parts of the discussion, will be restructured following comments by both reviewers and we will carefully harmonize and correct the terminology (i.e., Levantine Basin instead of eastern Mediterranean Sea).

2. Do the reconstructions indicate bottom or pore water oxygenation conditions? It is necessary to clarify whether the geochemical signals represent bottom or pore water conditions. Based on the low d13C values of authigenic carbonates from the HL layers during S1 deposition period, the authors proposed anoxic bottom water condition at site P33 (740 m water depth) and possibly at site P73 (570 m water depth). These sites are much shallower than the previously reported anoxic sites during S1 (1000 m and 2000m) based on authigenic carbonates (Aloisi et al., 2002; Bayon et al., 2013). It is not clear what the authors observed is pore water or water column oxygenation state. It is possible that the reconstructed oxygenation state was very localized with patchy distribution that is not suitable to generalize the whole basin.
About d18O values of LL2 carbonates, the authors proposed that "temperature and salinity gradients were homogenous throughout the water column". Does it mean that no density gradient existed during authigenic carbonate precipitation in 740 m water column? Is the hypothesis of the homogeneous water column consistent with oxygen depleted bottom water? If so, why the shallowest core (P99) from 400 m water depth showed less lamination despite and higher or comparable alkenone flux than at the deeper core (P33, Fig. 7)?

The point raised here by Reviewer #2 is an important one indeed. We propose here that the formation of the authigenic calcite layers results from processes occurring at the seafloor or at the sediment-water interface and reflects bottom-water chemistry since similar layers with an identical seasonal succession were found at locations 100 km apart from each other. These thin and distinct carbonate layers formed regularly in the sequence and are interlayered between other seasonal deposits, without any disturbance of the sedimentary fabric (esp. the soft LL2 layers). We therefore suggest that these carbonates formed during seasonally-occurring bottom-water supersaturation states. We ruled out a diagenetic origin of these layers resulting from chemical processes occurring deeper in the sediment column (i.e., with an isotopic signal reflecting pore-water chemistry) since such processes are often associated with local non-steady state diagenesis and are generally not found at nearby core sites. Only larger regional sedimentary disturbances (such as a turbidites or slumps) have been shown to lead to the formation of diagenetic carbonates (often deeper in the sediment column close to the sulfate-methane transition zone) that were found at several core sites (Kasten and Jörgensen, 2006). The precipitation of authigenic carbonates at cold seeps located deeper on the Nile DSF (Aloisi et al., 2002; Bayon et al., 2013) shows that anoxic conditions favoured carbonate precipitation over oxidation and degassing of CO2. Not mentioned in the paper, personal conversations with Dr. M. Revel about core MS27PT suggest that the thin seasonal carbonate layers were found at deeper sites MS27PT (1500m). We, however, cannot transfer our geochemical reconstruction to the entire basin because the observed layers may indeed only reflect local chemical conditions occurring under the influence of the Nile runoff. However, they provide a window of opportunity to assess seasonal changes in water chemistry based on sediments in which classical markers (benthic fauna) are missing. Future work will be focused on carrying out a systematic isotopic analysis of these sub-mm layers.
Concerning the second part of the comment about d18O values and water-mass structure: It is very difficult to interpret the pilot d18O values on authigenic calcite layers because we do not have any

constraints on bottom water d18O values or water temperatures during S1 sapropel deposition. However, we will add the figure below to the revised version of the manuscript in order to better visualise the changes in water-mass structure derived from our data. We can compare the biogenic and authigenic (inorganic) calcite isotopes because the offset is generally on the order of 0.1 permil (Bemis et al. 1998) and therefore much lower than the variations observed. This new figure shows that for some LL2 layers deposited in the older parts of the sapropel, the d18O is very similar to that of planktonic foraminifera, whereas LL3 layers show a d18O value similar to present-day benthic foraminifera. This suggests drastic reorganisation of the water-mass structure in the course of sapropel deposition. Our data indicate that at the beginning of S1, the water-column was largely influenced by isotopically lighter water, potentially deriving from large Nile freshwater input (with a dominant Blue Nile signature). With these few pilot samples, it is impossible to derive any detailed conclusions regarding water-mass structure and deoxygenation. Regarding P99, the depth of this core is closer to the mixed layer and therefore likely receives larger supply of oxygen from overlying water-masses. Therefore, anoxia would not develop even with a large carbon input.

[Figure]

At last, I have a comment on the general structure of the manuscript. The authors mixed result presentation and some interpretation in section 4 then more detailed discussion followed in section 5. Personally, I prefer to separate result and discussion to avoid mixed objective and subjective descriptions and redundant statements. I understand that this organization is due to multi-proxy reconstructions that necessitate to explain the meaning of numerous proxies. To overcome this complication, the authors may present the tool box classified into target variables before the result section.

Thank you for this comment. This was a point of discussion between co-authors as well for the reasons mentioned by the reviewer (mostly readability issues). We will follow the judicious advice of Reviewer #2 in the revised manuscript and restructure to separate results and interpretation. We will also provide a reading grid for the proxies.

I recommend to accept this work after major revision.

Minor / specific comments

Line 28 "the entire Levantine Basin". See my comment 1.

Will be modified accordingly.

Line 48 "decreased" should be replaced by "increased".

Done

Lines 51-52. " the consequently low or quasi-absent primary productivity (Krom et al.,2014)". Delete "or quasi-absent". Pujo-Pay et al. (2011) is probably more appropriate to cite.

Will be modified accordingly

Line 59. "the 1980's" should be "the 1980s".

Will be modified accordingly

Line 67. Somot et al., 2018. This paper treats the circulation in the NW Mediterranean Sea, not in the Eastern Mediterranean Sea. Adolff et al. (2015) is more suitable to this sentence because the circulation of the whole Mediterranean Sea is studied.

Will be modified accordingly

Line 72. "Sapropels have proven a valuable laboratory". This sentence is strange and should be revised.

This sentence will be modified accordingly.

Line 186. "Aavatech" should be "Avaatech".

Thank you for spotting the mistake, it will be corrected.

Line 218. "trialkyl" or "dialkyl"?

The C46 is a trialkyl tetraether lipid and so trialkyl is correct, however the acronym following this should read GTGT. This is now corrected.

Lines 204-212 and Fig. 4. Indicate the core name of which stable isotope data were obtained.

All stable isotope data were obtained on core P33 and this information will be added to section 3.6, table S4 and Fig. 4.

Lines 282-284. The occurrence of HL in both cores P33 and P73. Is the appearance of HL synchronous for the two cores to support the hypothesis of reduced oxygenation state at intermediate water depths?

This is a good point but unfortunately impossible to answer at present. We plan to count and measure layer thickness of core P73, too, which will help to align records and assess the synchronicity of these layers. Radiocarbon dating on such rapidly accumulating sediments is unfortunately too imprecise to tackle this question.

Line 367. Replace "contents" by "fluxes".

Will be modified accordingly.

Line 375. Add the corresponding water depths of the indicated cores after "deeper cores MS27PT and GeoB7702-3".

Indeed this information will be added.

Line 385. Add "variability" after "runoff".

Will be modified accordingly.

Line 400 and Fig. 8a,e,h. Why Ti-enrichment occurred at the Late Holocene?

This is a good question that has intrigued the authors for a while. A potential response to this question was provided in Blanchet et al. EPSL 2013. It is related to the fact that Ti/Ca traces relative variations between terrigenous and marine-derived sediment inputs and not purely terrigenous fluxes. So higher Ti/Ca ratios during the Late Holocene indicate higher terrigenous versus marine-derived sediment input, which was probably related to a re-establishment of the annual Nile floods after a very dry period at ca. 4 ka BP. However, the sediment input linked to monsoonal runoff during the Late Holocene was distinct from the situation during the early Holocene with regard to sediment budgets (sedimentation rates differing by two orders of magnitude – see Fig. 2). So, both periods are dominated by fluvial sediment input but the early Holocene was a period of massive fluvial erosion (and the core site was a depocenter for fluvial sediments).

Line 405. Add "ka BP" after "7.2".

Will be modified accordingly.

Line 406. "Ti/K" should be "K/Ti".

Will be modified accordingly.

Line 409 and Fig. 8. Indicate the sapropel interval in Fig. 8.

We will re-design Fig. 8 and add an indication of the sapropel layer.

Lines 436-438 and Fig. 3c. It is not clear that smectite, plagioclase and iron-titanium oxides are more abundant in DL2 relative to the other layers. According to the figure caption, Fe-rich phases are probably pyrite, not iron-titanium oxides.

The presence of Fe-Ti oxides is shown in Fig. 3b and if indeed pyrite represents a large fraction of the Fe-rich phases, this iron sulphide cannot account for the Ti-rich phases. Smectite and plagioclase (and to a smaller extent Fe-oxides) are found in all sediments analysed (albeit sometimes blurred by high CaCO3 contents), which indicates that detrital sediments mostly originated from volcanic outcrops. This was already shown before (e.g. publications by Eduardo Garzanti). In the text, we explicitly do not say that these layers are enriched in these minerals but only that they are present. We rather suggest that the background detrital mineral assemblage is quite similar between DL and LL2 layers (e.g., lines 459-460) and that dilution by carbonates controls their relative abundances. Obviously, this is still unclear and we will therefore rephrase this part of the text to improve clarity.

Line 459. Add a reference after "interface".

There is no reference for this sentence, this is a suggestion derived from our observations. ("The internal draping of these layers suggests…").

Lines 473-475. The authors may cite a recent regional modelling study that simulated surface salinity anomaly distribution in the Mediterranean Sea by increasing Nile river discharge (Vadsaria et al., 2019).

Very interesting paper, thanks for the suggestion!

Lines 476-479. It is unclear whether the authors treated here the mixing of water masses or sedimentation processes. Please clarify.

Thanks. We understand that the text is unclear and we will change it for clarification. The text does actually refer to both water-mass mixing and sedimentation processes. Double-diffusion is a typical process driving sedimentation and water-mass mixing in river deltas where low salinity, sediment-laden plumes enter a more saline water body. Salinity mixing and downward sediment transport is driven by the formation of salt fingers below the freshwater plume and supports the deposition of sediments on the seafloor (Parson et al. 2001; Jazi and Wells, 2016).

Lines 495-496. Add the reference for the bottom water d13C value of -7‰ off the Nile river mouth.

This value comes from our data (so there is no reference) but we will rewrite that part of the text for clarification.

Lines 511-514. About d18O of authigenic carbonates. What does "theoretical d18O" mean? D18O value of equilibrated calcite? If so, how do you estimate temperature and seawater d18O of surface and bottom waters?

Yes, theoretical d18O refers to inorganic calcite in equilibrium with seawater. To calculate the d18O of inorganic calcite, we have used
-   seawater temperature profiles at present from Medatlas (MEDAR Group, 2002)
-   sea surface temperatures for sapropel S1: median from Alkenone SST estimations (Blanchet et al. 2014, supplementary information)
-   seawater d18O profiles from Gat et al. (1996)
-   sea surface d18O for sapropel S1: median from d18Osw from Blanchet et al., 2014)
The equations used for calculating the equilibrium d18O of inorganic calcite and biogenic carbonates shown in Fig. 4 were adopted from Bemis et al (1998), Kim and O'Neil (1997) and Cornuault et al. ( 2018)

Lines 550-564. Delete the description corresponding to K. The results are already shown and K/Ti is not presented in Fig.9a.

Will be modified accordingly.

Lines 638-641. SIW was already defined.

Will be modified accordingly.

Numbering of figures. The authors used (a), (b), (c)...to describe the curves of re-construction instead labelling different panels (ex. Figs. 6, 7 and 8). Sometimes the different curves are combined and a common label is used. The lamination patterns are also labelled. This presentation is confusing. The authors may label the different panels or number all curves. The lamination patterns can be shown without numbering.

Well noted, we will modify the numbering accordingly. The figures will be corrected following Reviewer #2'comments and suggestions (Fig. 3, 5a,b and 8 caption).

Fig. 3. Please indicate which results correspond to which core (core P33 or core P73).

Fig. 5a, b. The x, y- axis and axis titles are too small.

Fig. 8 caption. There are errors of numbering. Please check it.

Fig. 9. A section map showing the site positions with the bathymetry will be helpful.

Good idea, thank you. It will be added to figure 9.

Fig. 10. Use different symbols to distinguish proxies used to evaluate the oxygenation states. Does each panel show the mean state of each period? What is the age rage of the interruption? Did the authors recalibrate the age of the different cores using the same calibration?

In accordance with comments from Reviewer #1, this figure will be removed as it blurs the main message.

---

## Author Response (AR1)

Response to reviewers

We are very grateful to both anonymous reviewers for the time they spend on our paper, for their constructive criticism and their comments. Both reviewers agree that the novelty and main interest of our paper lies in the seasonal record provided by these unique sediment cores. In this iteration, we have therefore focused the paper on these results rather than on drawing basin-wide conclusions. We hereafter reply to specific comments (in blue).

All changes are highlighted in the tracked-change version of the manuscript and we summarize here the main modifications:
1) Information on deep water formation added to 2. Regional context
2) Section 4. Results separated from the interpretations (now in section 5. Interpretation and discussion) and table 3 added to help for understanding the proxies
3) Structure of section 5 modified: better description of seasonal dynamics and interpretations from isotopes measured in LL2 (authigenic calcite)
4) Interpretation of millennial-scale records restricted to cores located in the SE Levantine Basin and at intermediate depth, better integration of modelling and proxy records.
5) Modification of figures 4-11 and removal of former Fig. 10 (Levantine Basin changes in oxygenation), addition of new Fig. 10.

Anonymous Referee #1

The authors present high-resolution sedimentological (microfacies) and geochemical data records for three upper bathyal sediment cores from high-accumulation sites off-shore the western distributaries of the Nile River delta. The sediment cores encompass the Holocene period including laminated sediments, which have been deposited during the last stagnation period of sapropel S1 in the Eastern Mediterranean Sea (EMS). The temporal resolution of the study region is unprecedented and allows for resolving even seasonal changes and thus the characterization of changes in the strength of the yearly Nile flood and associated regional environmental conditions at the core sites. In this reconstruction, the application of microfacies analyses, previously established for laminated lake sediments provides essentially novel information on the paleohydrology of the Nile River and related summer floods. The manuscript is generally well designed and follows a widely comprehensive argumentation.

Thank you for these supportive comments.

Nevertheless, the following issues should be considered in the revision process:

1) I disagree with the general conclusion concerning the role of productivity as a major driver in S1 formation in the entire EMS basin. Enhanced productivity likely plays an important role during S1 deposition in the vicinity of the Nile delta and in the eastward flowing surface water current along the Israeli coast. Eutrophication of surface waters has been documented in sediments retrieved from the Israeli continental margin. However, evidence from abyssal and bathyal core sites from the central parts of the basins and from the northern margins of the EMS provide a controversial picture, in many cases lacking evidence for eutrophication during S1 formation. In this context, the manuscript would profit from a more critical discussion of preservation issues with respect to the interpretation of geochemical proxies since syn- or post-depositional degradation of organic compounds may play a significant role. For example, the study of Moebius et al. (2010, Biogeosciences 7, 3901-3914) suggested that "... preservation plays a major role for the accumulation of organic-rich sediments casting doubt on the need of enhanced primary production

for sapropel formation." In addition, the study of Grimm et al. (2015) used a sophisticated regional ocean circulation model coupled to a biogeochemical model to explore the potential role of productivity and associated organic carbon fluxes in S1 formation (this is exactly what you ask for at the end of your conclusions!). They found that the required oxygen depletion is mainly depending on the stratification intensity and stagnation history of the water column and that a basin-wide productivity increase is not required as a prerequisite for sapropel formation. These and other results suggest that the observations of upper bathyal anoxia off the Nile delta may not be representative for environmental changes in the entire basin but may rather represent a regional situation due to its vicinity to the Nile nutrient source. This suggestion is also confirmed by the restriction of sustained basin-wide early Holocene anoxia to water depths below approximately 1500 m (De Lange et al.,2008, Nature Geoscience, 1: 606–610) while sustained anoxia seem to have prevailed at shallower water depths under the direct influence of Nile outflow waters. In the revision of the manuscript, the authors should acknowledge the still controversial findings and diverse evidence from both proxy and model applications. In doing so, I suggest to abandon conclusions drawn for the entire EMS (including Figure 10) and rather focus on the Nile river dynamics and regional environmental impacts. These results are novel enough and provide a variety of essentially new insights into past hydrological changes and marine depositional responses.

We thank Reviewer #1 for this important comment, which helped to separate processes occurring in deeper parts of the Levantine Basin and those occurring at intermediate depth. The interplay between productivity, preservation and stagnation is a complex issue and we acknowledge that it is probably wiser to draw conclusions for the environments "closer to home", i.e., directly under the influence of the Nile sediment plume.
- Preservation: Reviewer #1 will find in §5.3 (lines 537-541 and 546-551) a discussion on preservation issues.
- Stagnation and eutrophication: In §5.3.1 to §5.3.3, we have focused the discussion on comparing our records to sites located further downstream of the Levantine Jet, and those at intermediate depth (<1800 m). Former Fig. 10 was removed and we modified new Fig. 11 (former Fig. 9) to improve readability.

2) You argue that the Nile flood triggered blooms of planktonic foraminifera and calcareous nannoplankton in autumn, similar to natural Nile blooms prior to the construction of the Aswan dam. Typically, eutrophication in surface waters influenced by riverine nutrient input results in diatom blooms as described for the historical Nile floods (Halim et al., 1967). To date, Halim et al. (1967) do not mention planktonic foraminifera and coccolithophorids as typical groups responding to the seasonal nutrient input. In the modern EMS, planktonic foraminifera and coccolithophorids are widely distributed and even thrive in the ultraoligotrophic parts of the basin as in many other oligotrophic oceans. Did you observe any layers rich in opal? In the EMS, opal remains are often not pre-served, which may inhibit documentation of the proper succession of nutrient-driven phyto- and zooplankton association in the sediment. Your discussion and interpretation should be more specific here, acknowledging a correct assessment of plankton response to river-induced surface water eutrophication.

Reviewer #1 is correct: Halim et al. (1967) did not specifically report carbonate zooplankton for historical floods of the Nile. We have revised section 5.1 as suggested by both reviewers: we indicate that foraminifera and coccolith remains are associated to remains of organic matter (see Supplementary Fig. S3), we acknowledge the absence of diatoms/opal probably due to preservation issues (lines 429-430). We also discussed the abundance of *G. ruber* in this flood-derived fresher water (lines 418-425).

3) The presence of benthic foraminifera (BF) in the interruption and upper part of S1clearly indicates the absence of permanent anoxic conditions and at least intermittently oxygenated time periods at the deepest core site (738 m water depth) since around8200 years B.P. It is a pity that no BF data are available for the other two cores. On the other hand, the presented data on the genus level do not provide sufficient insights into the BF fauna and thus benthic ecosystem state since different species of the same genus often have contrasting ecology. In addition, proper taxonomy is required for excluding the potential effects of down-slope transport. The authors mention the genus Cibbides(?). I guess you refer to the genus Cibicides or Cibicidoides, do you? Species of the genus Cibicides commonly inhabit shelf environments in the EMS and thus would represent reworked tests when found on the continental slope. On the other hand, Cibicidoides is an autochthonous deep-sea taxon. In order to avoid misinterpretations, it would be more honest to lump all species together and present data on the BF presence/absence or concentration (e.g. individuals per g dry sediment).

As suggested by Reviewer #1, we have combined the specimens of different genera and show the BF data as individuals/g sediment (Fig. 6c and 11b).

4) I wonder if you have observed any gypsum crystals in the sediment. Post-depositional oxidation of iron sulfides and calcium carbonate often result in precipitation of gypsum in sapropelic sediments.

No gypsum was observed in this core, which might be due to the relatively low Corg content (±1%) compared to other sapropel layers (our cores rather represent sapropelites *sensu stricto*).

5) Some of the figures appear too busy and the blueish colors used do not contrast sufficiently. This is particularly visible in figures 6, 8 and 9. I suggest using more contrasting colors and also avoid overlying symbols and lines.

We have modified Fig. 6, 7 and 8, as well as former Fig. 9 (now labelled 11) and hope that they are more readable now. All biomarker results are now presented in Fig. S3.

I hope that my suggestions prove useful for the revision of the manuscript!

Yes, they are, thank you very much!

Anonymous Referee #2

Applying multiproxy approach (micro-facies analysis with inorganic and organic geo-chemistry) to three cores of high sedimentation rates from the Nile deep-sea fan, Blanchet et al. reconstructed seasonal to millennial-scale variability of detrital inputs, biological productivity and bottom/pore water oxygenation states during the Holocene covering sapropel S1 deposition period. The highly resolved records provide very useful information on the impact of Nile flooding as both freshwater and nutrient supplier. The quality of data is high and scientific subject fits well the field covered in Climate of the Past. I, however, have several concerns that should be solved during the revision process. I develop the points below.

Thank you for this positive assessment.

1. Local vs. regional impact of Nile discharge: circulation and productivity The most innovative results of the study are (i) annual cycle of sediment deposition during S1period in the western Nile deep-sea fan and (ii) additional evidence for heterogeneous bottom/pore water oxygenation states in the coastal regions during S1. Point (i) is well described whereas the transition from (i) to (ii) is abrupt and some key points of (ii) are missing. The most exciting but not fully addressed subject is how the identified annual cycle of Nile discharge and the productivity changes was interacted with basin-scale water circulation and oxygenation states. The authors briefly proposed the impact of nutrient supply by Nile river discharge in relation to the mega-summer monsoon of the AHP by analog with "Nile bloom". However, there exists a growing body of evidences for the leading role of stagnant circulation that pre-conditioned the S1 deposition based on proxy reconstruction and a numerical modeling (Grimm et al., 2015). Since the authors provided unpreceded resolution data, it could be possible to revisit the role of the Nile at finer timescale. For instance, the authors proposed that fine-grained clay-rich particle deposition in laminated layers as a sign of stagnant circulation. How the stagnant circulation was realized and what was the relationship between the observed slower circulation in the Nile deep-sea fan with the ventilation of the other parts of the Levantine Sea? In general, the present manuscript did not sufficiently describe the role of the circulation to the sapropel formation. The present-day circulation pattern is very shortly shown in Section 2 (Regional context) and there is no statement about water mass occupies the three core sites and how the mass is oxygenated under the present condition. This information should be added. Besides, the sketches of an annual cycle shown in Fig. 5c are difficult to understand. What is the size of geographical extension of area for the proposed sedimentation processes?

To clarify the local vs. regional impact of Nile discharge, I would suggest reorganize section 5.3. Do the titles indicated on lines 574, 601 and 653 correspond to the subsection of 5.3, thus the section extends for 7 pages? It is better to start by coastal regions (Nile deep sea fan and Israeli coast) and extend the discussion to the Levantine basin. The authors should be careful about the terms of geographic definition. For instance, the whole Eastern Mediterranean Sea, including the Adriatic Sea and Ionian Sea, is not treated in the present manuscript, therefore it is inappropriate to use this term in the section title.

We thank Reviewer #2 for these useful comments.
- Articulation between 5.1, 5.2 and 5.3: we have rewritten and re-organized these sections in order to better show how seasonal-scale findings help to draw conclusions on millennial-scales.
- Seasonal-scale results: we have revised parts 5.1 and 5.2 in order to better highlight the potential of our results for understanding the interplay between stagnation and river runoff.
- Deposition of clay-sized minerals during the winter: we have not further developed this part because the deposition of clays might indeed indicate seawater stagnation but can also be

enhanced by processes such as flocculation, which increases the settling speed of fine minerals. However, we discuss the persistence of a low-flow regime in winter with regard to the potential to form intermediate waters in the Aegean Sea (lines 440-448).
- Section 2 (Regional context) was revised to incorporate information on present-day intermediate and deep-water masses. We also discuss how our new data feed into the discussion on water-mass distribution during S1 deposition (see previous point)
- Structure of the discussion (§ 5): it has been modified to incorporate comments from both reviewers. We discuss our results in the context of the Levantine Jet and the intermediate water masses rather than at the scale of the whole Levantine Basin.
- Mentions to geographical locations have been harmonized to SE Levantine Basin and Nile DSF.

2. Do the reconstructions indicate bottom or pore water oxygenation conditions? It is necessary to clarify whether the geochemical signals represent bottom or pore water conditions. Based on the low d13C values of authigenic carbonates from the HL layers during S1 deposition period, the authors proposed anoxic bottom water condition at site P33 (740 m water depth) and possibly at site P73 (570 m water depth). These sites are much shallower than the previously reported anoxic sites during S1 (1000 m and 2000m) based on authigenic carbonates (Aloisi et al., 2002; Bayon et al., 2013). It is not clear what the authors observed is pore water or water column oxygenation state. It is possible that the reconstructed oxygenation state was very localized with patchy distribution that is not suitable to generalize the whole basin.
About d18O values of LL2 carbonates, the authors proposed that "temperature and salinity gradients were homogenous throughout the water column". Does it mean that no density gradient existed during authigenic carbonate precipitation in 740 m water column? Is the hypothesis of the homogeneous water column consistent with oxygen depleted bottom water? If so, why the shallowest core (P99) from 400 m water depth showed less lamination despite and higher or comparable alkenone flux than at the deeper core (P33, Fig. 7)?

Large parts of sections 5.1 and 5.2 have been revised.
- Authigenic or diagenetic calcite, i.e., bottom-water or pore-water signal: this is discussed in § 5.1, lines 449-460. We propose several lines of evidence supporting that soft LL2 formed at the seafloor (authigenic calcite formed from bottom waters) while the formation process of cemented HL is not clearly determined.
- We separate the discussion on bottom-water chemistry based on $\delta^{13}C$ values (§ 5.2.1) from that on water-column stratification and precipitation of authigenic calcite mostly based on $\delta^{18}O$ values (§ 5.2.2). We also added Fig. 10 showing the stratification of the water-column for different LL2.
- We propose that soft LL2 formed at the beginning of the flood season, when cation-rich freshwater interacted with anoxic, bicarbonate-rich bottom waters. Some LL2 have d18O signatures similar to those of planktonic foraminifera and we propose that efficient water mixing due to the formation of salt fingers drove the low-salinity flood plume to deeper parts of the Nile DSF (see discussion in § 5.2.2).
- Shallower core P99 being closer to the mixed layer, it might have ventilated from the top of the water-column and therefore remained suboxic even with strong fluxes of organic matter.

At last, I have a comment on the general structure of the manuscript. The authors mixed result presentation and some interpretation in section 4 then more detailed discussion followed in section 5. Personally, I prefer to separate result and discussion to avoid mixed objective and subjective descriptions and redundant statements. I understand that this organization is due to multi-proxy reconstructions that necessitate to explain the meaning of numerous proxies. To overcome this

complication, the authors may present the tool box classified into target variables before the result section.

Following Reviewer #2's advice, we have separate results (§ 4) from interpretations and discussion (§ 5) and added Table 3 to provide a reading grid.

I recommend to accept this work after major revision.

Minor / specific comments

Line 28 "the entire Levantine Basin". See my comment 1.

Modified throughout the manuscript.

Line 48 "decreased" should be replaced by "increased".

Done

Lines 51-52. " the consequently low or quasi-absent primary productivity (Krom et al.,2014)". Delete "or quasi-absent". Pujo-Pay et al. (2011) is probably more appropriate to cite.

Done

Line 59. "the 1980's" should be "the 1980s".

Done

Line 67. Somot et al., 2018. This paper treats the circulation in the NW Mediterranean Sea, not in the Eastern Mediterranean Sea. Adolff et al. (2015) is more suitable to this sentence because the circulation of the whole Mediterranean Sea is studied.

Done

Line 72. "Sapropels have proven a valuable laboratory". This sentence is strange and should be revised.

Done, it now reads "Sapropels provide a natural laboratory…" (l. 79)

Line 186. "Aavatech" should be "Avaatech".

Thank you for spotting the mistake, it is corrected.

Line 218. "trialkyl" or "dialkyl"?

The C46 is a trialkyl tetraether lipid and so trialkyl is correct, however the acronym following this should read GTGT (it has been removed). This is now corrected.

Lines 204-212 and Fig. 4. Indicate the core name of which stable isotope data were obtained.

All stable isotope data were obtained on core P33 and this information has been added to section 3.6, table S4 and Fig. 4, 5 and 10 (and their captions).

Lines 282-284. The occurrence of HL in both cores P33 and P73. Is the appearance of HL synchronous for the two cores to support the hypothesis of reduced oxygenation state at intermediate water depths?

This is a good point but unfortunately impossible to answer at present. We plan to count and measure layer thickness of core P73, too, which will help to align records and assess the synchronicity of these layers. Radiocarbon dating on such rapidly accumulating sediments is unfortunately too imprecise to tackle this question.

Line 367. Replace "contents" by "fluxes".

Done

Line 375. Add the corresponding water depths of the indicated cores after "deeper cores MS27PT and GeoB7702-3".

Done

Line 385. Add "variability" after "runoff".

Sentence removed

Line 400 and Fig. 8a,e,h. Why Ti-enrichment occurred at the Late Holocene?

This is a good question that has intrigued the authors for a while. A potential response to this question was provided in Blanchet et al. EPSL 2013. It is related to the fact that Ti/Ca traces relative variations between terrigenous and marine-derived sediment inputs and not purely terrigenous fluxes. So higher Ti/Ca ratios during the Late Holocene indicate higher terrigenous versus marine-derived sediment input, which was probably related to a re-establishment of the annual Nile floods after a very dry period at ca. 4 ka BP. However, the sediment input linked to monsoonal runoff during the Late Holocene was distinct from the situation during the early Holocene with regard to sediment budgets (sedimentation rates differing by two orders of magnitude – see Fig. 2). So, both periods are dominated by fluvial sediment input but the early Holocene was a period of massive fluvial erosion (and the core site was a depocenter for fluvial sediments).

Line 405. Add "ka BP" after "7.2".

Done

Line 406. "Ti/K" should be "K/Ti".

No, it is Ti/K throughout the text and in figures.

Line 409 and Fig. 8. Indicate the sapropel interval in Fig. 8.

We removed the mention of the sapropel and therefore did not modify Fig. 8.

Lines 436-438 and Fig. 3c. It is not clear that smectite, plagioclase and iron-titanium oxides are more abundant in DL2 relative to the other layers. According to the figure caption, Fe-rich phases are probably pyrite, not iron-titanium oxides.

This sentence now reads (Lines 402-405): "The presence of silt-sized particles of smectite, plagioclases and iron-titanium oxides in the DL2 matches the grain size of particulate matter transported during peak Nile discharge (Billi and el Badri Ali, 2010), as well as its mineralogical composition, typical for basaltic rocks of the Ethiopian volcanic plateau (Garzanti et al., 2015, 2018) (Fig. 3c)."

Line 459. Add a reference after "interface".

There is no reference for this sentence, this is a suggestion derived from our observations (lines 452-455).

Lines 473-475. The authors may cite a recent regional modelling study that simulated surface salinity anomaly distribution in the Mediterranean Sea by increasing Nile river discharge (Vadsaria et al., 2019).

Very interesting paper, thanks for the suggestion! Paper cited repeatedly in the revised manuscript.

Lines 476-479. It is unclear whether the authors treated here the mixing of water masses or sedimentation processes. Please clarify.

This part has been rewritten and is hopefully clearer, see lines 405 to 413.

Lines 495-496. Add the reference for the bottom water d13C value of -7‰ off the Nile river mouth.

This value comes from our data (so there is no reference) but we this part has been revised (see § 5.2.1, lines 470-471.

Lines 511-514. About d18O of authigenic carbonates. What does "theoretical d18O" mean? D18O value of equilibrated calcite? If so, how do you estimate temperature and seawater d18O of surface and bottom waters?

Yes, theoretical $\delta^{18}O$ refers to inorganic calcite in equilibrium with seawater. This has been modified throughout the text and in figures 5 and 10.
All necessary information to calculate these $\delta^{18}O$ values are now given in the Supplementary information and in Table S4.

Lines 550-564. Delete the description corresponding to K. The results are already shown and K/Ti is not presented in Fig.9a.

Removed.

Lines 638-641. SIW was already defined.

Modified accordingly.

Numbering of figures. The authors used (a), (b), (c)...to describe the curves of re-construction instead labelling different panels (ex. Figs. 6, 7 and 8). Sometimes the different curves are combined and a common label is used. The lamination patterns are also labelled. This presentation is confusing. The authors may label the different panels or number all curves. The lamination patterns can be shown without numbering.

Numbering has been modified accordingly for Fig. 6,7,8.

Fig. 3. Please indicate which results correspond to which core (core P33 or core P73).

All results are from core P33, indicated in revised Figure 3

Fig. 5a, b. The x, y- axis and axis titles are too small.

This figure has been split and the axis have been modified (revised Fig. 4).

Fig. 8 caption. There are errors of numbering. Please check it.

Caption corrected.

Fig. 9. A section map showing the site positions with the bathymetry will be helpful.

Good idea, thank you. A map has been added to revised Fig. 11.

Fig. 10. Use different symbols to distinguish proxies used to evaluate the oxygenation states. Does each panel show the mean state of each period? What is the age rage of the interruption? Did the authors recalibrate the age of the different cores using the same calibration?

In accordance with comments from Reviewer #1, this figure will be removed as it blurs the main message.

References All added to the revised manuscript

Adloff, F., Somot, S., Sevault, F., Jordà, G., Aznar, R., Déqué, M., Herrmann, M., Marcos, M., Dubois, C., Padorno, E., Alvarez-Fanjul, E., and Gomis, D.: Mediterranean Sea response to climate change in an ensemble of twenty first century scenarios, Clim. Dyn., doi: 10.1007/s00382-015-2507-3, 2015. 1-28, 2015.

Pujo-Pay, M., Conan, P., Oriol, L., Cornet-Barthaux, V., Falco, C., Ghiglione, J. F.,Goyet, C., Moutin, T., and Prieur, L.: Integrated survey of elemental stoichiometry (C,N, P) from the western to eastern Mediterranean Sea, Biogeosciences, 8, 883-899,2011.

Vadsaria, T., Ramstein, G., Dutay, J. C., Li, L., Ayache, M., and Richon, C.: Simulating the Occurrence of the Last Sapropel Event (S1): Mediterranean Basin Ocean Dynamics Simulations Using Nd Isotopic Composition Modeling, Paleoceanography and Paleoclimatology, 34, 237-251, 2019.